# Graph-structured populations elucidate the role of deleterious mutations in long-term evolution

Nikhil Sharma [1] ✉, Suman G. Das[2,3], Joachim Krug[1,4] & Arne Traulsen [1] ✉

Birth-death models are used to understand the interplay of genetic drift and natural selection. While well-mixed populations remain unaffected by the order of birth and death and where selection acts, evolutionary outcomes in spatially structured populations are affected by these choices. We show that the choice of individual moving to vacant sites—parent or offspring—controls the initial mutant placement on a graph and hence alters its fixation probability. Moving parent individuals introduces, to our knowledge, previously unexplored update rules and fixation categories for heterogeneous graphs. We identify a class of graphs, amplifiers of fixation, where fixation probability is larger than in well-mixed populations, regardless of the mutant fitness. Under death-Birth parent moving, the star graph is an amplifier of fixation, with a non-zero fixation probability for deleterious mutants, in contrast to very large well-mixed populations. Most Erdős-Rényi graphs of size 8 are amplifiers of fixation under death-Birth parent moving, but suppressors of fixation under Birth-death offspring moving. Surprisingly, amplifiers of fixation attain lower fitness in long-term evolution, despite favouring beneficial mutants, while suppressors of fixation attain higher fitness. These counterintuitive findings are explained by the fate of deleterious mutations and highlight the crucial role of deleterious mutants for adaptive evolution.

Population structure can substantially impact the evolution of a population[1–11]. Understanding the role of spatial structure in evolutionary biology is crucial and demands moving beyond the commonly assumed well-mixed populations[12]. Evolutionary graph theory provides a platform where structured populations are modelled as graphs[13], with each node representing an asexually reproducing individual and the links defining its interaction neighbourhood. In this framework, a complete graph represents a well-mixed population where each node interacts with every other node with equal propensity.

Fixation probability and fixation time are two key observables in evolutionary graph theory. The fixation probability of a mutant is the

probability for the mutant to take over a population of wild-types[14–18], while the (conditional) fixation time represents the duration it takes for this process to complete[19–22]. The fixation time of a mutant is a random variable with a specific distribution, and the quantity of interest generally is the average fixation time[23].

The fixation probability is an important quantity in evolutionary biology, because it determines the rate of evolution[24–26]. During a fixation event, two evolutionary forces are at play—natural selection and genetic drift—and spatial structure can modulate the strength of these forces[27]. With the well-mixed population serving as the reference, graph structures that amplify the strength of selection are termed amplifiers of selection (AoS), while those that suppress it are

[1]Department of Theoretical Biology, Max Planck Institute for Evolutionary Biology, Plön, Germany. [2]Institut für Ökologie und Evolution, Universität Bern, Bern, Switzerland. [3]Swiss Institute of Bioinformatics, Lausanne, Switzerland. [4]Institute for Biological Physics, University of Cologne, Cologne, Germany. ✉e-mail: nsharma@evolbio.mpg.de; traulsen@evolbio.mpg.de

referred to as suppressors of selection (*SoS*)[13,15]. An *AoS* is a graph that has a higher probability of fixing beneficial mutants and a lower probability of fixing deleterious mutants compared to the complete graph. On the other hand, a *SoS* is a graph that has a lower probability of fixing beneficial mutants and a higher probability of fixing deleterious mutants.

In general, graphs can be weighted and directed[28–30]. This means that an individual may not interact with its neighbour as strongly as the neighbour interacts with the focal individual. In this work, we focus on unweighted and undirected graphs. The precise form of the interactions among individuals is determined by an update rule. The commonly studied update rules in evolutionary graph theory are the Moran Birth-death (Bd) and the Moran death-Birth (dB) updates[31–33]. The shorthand Bd implies that the birth event precedes the death event. The uppercase B indicates that selection operates during the birth event, while the lowercase d represents the neutral nature of the death event. This offers various choices for the birth-death updates[34]. References [35,36] show that for regular graphs the fixation probabilities under Db updating are equivalent to those under Bd, and the fixation probabilities under bD updating are equivalent to those under dB. However, for heterogenous graphs, this is not true[37]. In general, the fixation probability of a mutant on a graph depends crucially on the update rule[35]. In ref. [33] it was found that for Bd updating, most small random graphs are *AoS*, whereas, under dB updating, most of the small random graphs are *SoS*.

However, not only the update rule but also the node where the mutant initially appears substantially affects the fixation probability. Mutant initialisation schemes determine the likelihood for a node to be initialised with the mutant. Two popular schemes are uniform mutant initialisation and temperature mutant initialisation[38]. Under uniform mutant initialisation, every node is equally likely to be initialised with the mutant. For temperature initialisation, the initial mutant is more likely to appear on nodes with higher turnover rates, that is, with a larger number of links. The star graph is an *AoS* under Moran Bd updating with uniform mutant initialisation. However, for temperature initialisation, the star graph is a suppressor of fixation (*SoF*)—a graph with lower probability of fixing a mutant than the well-mixed population, regardless of the mutant's fitness.

Recently, some studies in evolutionary graph theory went beyond the fixation time scales and explored the state of mutation-selection balance that emerges at long times[39–41]. When the uniformly initialised star graph (an *AoS*) was subjected to long-term mutation-selection dynamics, it achieved a higher average steady-state fitness compared to the complete graph[42]. This outcome was anticipated because an *AoS* is more efficient at fixing beneficial mutants and preventing the fixation of deleterious mutants. Surprisingly, however, the temperature initialised star graph, despite being a *SoF*, not only attained a higher fitness than the complete graph, but also equally high fitness as the uniform initialised star graph. This result can be explained by the ability of the temperature initialised star graph to efficiently reject deleterious mutants, compensating for its inability to fix beneficial mutants.

To our knowledge, this is the only known example so far where the deleterious mutant regime has the potential to influence long-term evolution, and it raises a number of questions. How common is *SoF* for temperature initialised Bd updating? Do all of *SoF* attain higher fitness than the complete graph, despite having lower probabilities of fixing advantageous mutants? What about dB updating? Does the deleterious mutant regime play any significant role for dB long-term dynamics as well? We address all of these questions here.

The structure of this paper is as follows: We begin by establishing a connection between update rules and mutant initialisation schemes. We show that these schemes naturally arise from the choice of individuals moving to vacant nodes—either the parent-type offspring or the mutant offspring—and therefore do not need to be separately specified. Subsequently we study the star graph and Erdős-Rényi random graphs at short-term fixation time scales, considering various update rules. Notably, we observe that the star graph acts as an amplifier of fixation (*AoF*) under temperature initialised dB updating with a higher probability of fixing mutants compared to the complete graph, regardless of the fitness value. Similarly, we find that most of the small random graphs are *AoF* under temperature initialised dB updating and *SoF* under temperature initialised Bd updating. Additionally, we study the star graph and random graphs under long-term mutation-selection dynamics. Surprisingly, despite being *SoF*, most of the random graphs achieve higher fitness than the complete graph for temperature initialised Bd updating, whereas most of the random graphs attain lower fitness for temperature initialised dB updating despite being *AoF*.

## Results

### Update mechanisms in graph-structured populations

When working with structured populations, the update rule substantially affects the dynamics. An evolutionary update rule determines not only the order of the birth and death events and the choice of event(s) where selection operates but also the process by which new mutations appear in the population. While mutations can appear spontaneously, for example, due to stress such as exposition to UV radiation, in this work we consider the case when mutations are coupled to birth.

A mutant initialisation scheme $\mathcal{I}$ denotes the probability distribution with which an initial mutant appears on a node, $p = (p_0, p_1, \cdots p_{N-1})$. For example, in the uniform mutant initialisation scheme $\mathcal{U}$, we have $p_i = 1/N$ for all nodes $i$, i.e., a mutant appears in every node with the same probability. Similarly, for the temperature initialisation scheme $\mathcal{T}$, the probability for a node to receive an initial mutant is proportional to the temperature (sum of incoming/outgoing weight) of the node.

As an example for temperature initialisation, let us focus on the Moran Birth-death (Bd) update rule with the offspring moving to another site[38]. First, an individual is selected with probability proportional to its fitness to give birth to an offspring. Thus fitness is equivalent to the reproduction rate of an individual. After reproduction, the offspring either resembles its parent with probability $1 - \mu$ or is a mutant with probability $\mu$. Then the offspring takes over the node of a random neighbouring individual chosen to die. Therefore, in a population where every individual has the same fitness, the first mutant is more likely to appear on nodes with higher degree. To be specific, the probability $p_i$ that a mutant appears in a node $i$ is proportional to its in-temperature $\mathcal{T}_i^{\text{in}}$,

$$p_i = \frac{1}{N} \underbrace{\sum_j \frac{a_{ji}}{\sum_k a_{jk}}}_{\text{in-temp of node } i} = \frac{1}{N} \mathcal{T}_i^{\text{in}}. \tag{1}$$

Here, $a_{lm}$ is an element of the adjacency matrix **A** with value equal to 1 if there is link directed from node $l$ to $m$ and 0 otherwise. We work with undirected graphs for which **A** is a symmetric matrix. For further examples for this update mechanism, see also refs. [43,44].

So far, we have assumed that the offspring moves to a neighbouring node while the parent remains at its position, see Fig. 1A. To denote this, we use the shorthand Bd$^o$ where the superscript indicates that the offspring moves to a vacant site. With the same assumption, for death-Birth updating dB$^o$, the mutant initialisation is uniform: The probability that an initial mutant appears in node $i$ under dB$^o$ updating is equal to 1/$N$.

To explore other possibilities, we consider the case where the parent moves to the neighbouring node and the offspring stays at the original node, see Fig. 1B. These update rules will be denoted by the superscript $p$. Parent moving rules are relevant for microbial and

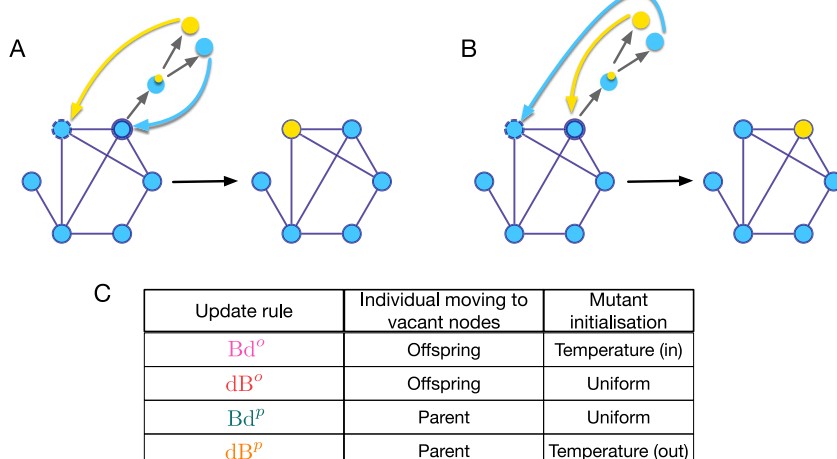

| Update rule | Individual moving to vacant nodes | Mutant initialisation |
|---|---|---|
| Bd$^o$ | Offspring | Temperature (in) |
| dB$^o$ | Offspring | Uniform |
| Bd$^p$ | Parent | Uniform |
| dB$^p$ | Parent | Temperature (out) |

**Fig. 1 | Type of individual moving to vacant sites and mutant initialisation schemes.** The mutant initialisation scheme, the likelihood that the initial mutant appears on a given node in a homogenous fitness background, is fully determined by the evolutionary update rule, provided mutations are coupled to reproduction and the choice of individual that moves to the vacant node is specified. This is shown in (**A, B**), where wild-type individuals are shown in blue and the mutant in yellow. The individual chosen for birth is marked by a thick solid circle while the one chosen for death is marked by a dashed circle. In case of mutation, one of the daughter cells is a mutant (yellow type) and the other one resembles the mother (blue type). In (**A**), the mutant offspring moves to the vacant site. Throughout the paper, this is referred to as the offspring moving rule. For Bd updating, the corresponding initialisation scheme Bd$^o$ is temperature initialised, and for dB updating, dB$^o$, it is uniformly initialised. Similarly, in (**B**) the parent-type offspring moves to the vacant node while the mutant offspring stays at the birth site. We call this the parent moving rule. The corresponding Bd update rule Bd$^p$ leads to uniform initialisation, whereas dB$^p$ implies temperature initialisation. **C** lists the combinations of update rules and the choices of the individual moving to the vacant site. In the rest of the paper, the following colour-coding has been used for the update rules: pink for Bd$^o$, red for dB$^o$, teal for Bd$^p$, and orange for dB$^p$.

somatic cell populations, where parent and offspring cannot necessarily be distinguished, see Fig. 1. In such cases, there is no a priori reason why e.g. a new daughter cell is leaving the mother cell's site and not vice-versa. There are numerous studies in the mathematical oncology literature that investigate mutation-selection dynamics on regular graphs[45,46]. In all of these studies, the mutant daughter cell moves to the vacant site, but there is no reason why the daughter cell identical to the mother cell cannot move instead. On regular graphs, the initial placement of the mutant is independent of the choice of individual moving to vacant site—parent (daughter cell resembling the mother) or offspring (mutant daughter cell). This is not true for non-regular graphs, where the initial placement of the mutant is determined by the choice of individual moving to vacant sites (Fig. 1). While we mostly focus on the extreme cases of offspring and parent movement, the intermediate case where either the offspring or the parent move with a certain probability is of natural interest and is covered in the Supplementary Note 2.

The probability that the initial mutant under dB$^p$ update arises in node $i$ is

$$p_i = \frac{1}{N} \underbrace{\sum_j \frac{a_{ij}}{\sum_k a_{kj}}}_{\text{out-temp of node } i} = \frac{1}{N} \mathcal{T}_i^{\text{out}}. \tag{2}$$

Thus, the mutants for dB$^p$ updating with parent-type offspring moving to a vacant node are out-temperature initialised. In other words, highly connected nodes are more likely to receive an initial mutant. For unweighted and undirected graphs, $\mathcal{T}^{\text{in}} = \mathcal{T}^{\text{out}}$.

In the literature, it has been suggested that for dB updating temperature initialisation does not exist[47,48]. This is true when the offspring individual moves to a neighbouring node. But when we instead assume that the parent moves, we obtain the temperature initialised dB update. Moreover, for Bd$^p$ updating with the parent moving to a vacant node, we recover the uniform mutant initialisation. Figure 1C lists the combination of update rules and the choice of

individual moving to the vacant site leading to different mutant initialisation schemes.

### Short time scales: fixation dynamics
In evolutionary graph theory, the focus is typically on the fixation probability of a mutant. For a given graph, this probability may depend on the node where the mutant first appears. To make this explicit, we denote by $\phi_{G,i}(f',f)$ the fixation probability of a mutant with fitness $f'$ in a wild-type population of fitness $f$ on a graph $G$ starting from node $i$. As discussed above, the mutant usually does not arise in every node with the same probability. For a general mutant initialisation scheme $\mathcal{I}$, the average fixation probability on graph $G$ is

$$\Phi_G^{\mathcal{I}}(f',f) = \sum_{i=0}^{N-1} p_i \cdot \phi_{G,i}(f',f). \tag{3}$$

For the case of well-mixed population, because of the symmetry of the graph the fixation probability is independent of the mutant initialisation scheme. This is true for all regular graphs. Under dB updating, the fixation probability for a mutant with fitness $f'$ in a population with fitness $f$ on the complete graph is[15,33]

$$\Phi_{\text{dB},C}^{\mathcal{I}}(f',f) = \Phi_{\text{dB},C}(f',f) = \frac{N-1}{N} \frac{1 - \frac{f}{f'}}{1 - \left(\frac{f}{f'}\right)^{N-1}}. \tag{4}$$

In the limit of large $N$, this becomes equal to the Moran Bd fixation probability for the complete graph[15]. In the following, we study average fixation probabilities on the star graph and on random graphs for different update rules.

**The star graph.** The star graph has been extensively studied in evolutionary graph theory, as it is highly inhomogeneous but still analytically tractable[49]. The Moran Bd process on the star graph was studied in[13]. Since then, the star graph has become the prime example of an *AoS* for uniform mutant initialisation, which can be solved exactly[50,51] for various

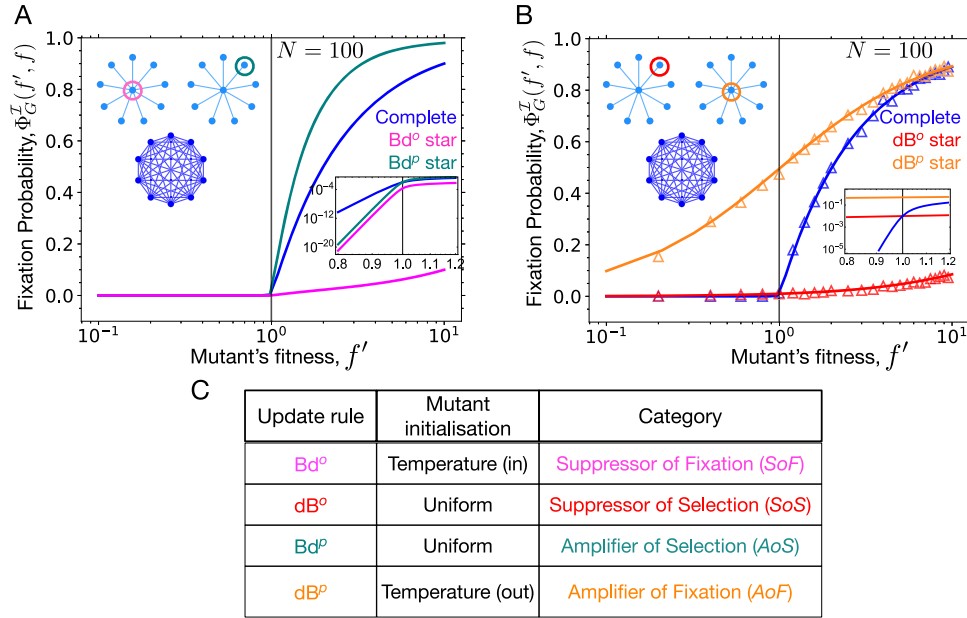

**Fig. 2 | Fixation probability for the star graph under different update rules.** For large population sizes, the offspring and parent update rules result in the initial mutant placement either on the centre or on a leaf of the star graph. Depending on the update rule (Bd or dB) and the type of individual moving, encircled nodes of the star graphs in (**A**, **B**) depict the most likely initial mutant placement. **A** Under temperature initialised Bd updating (Bd$^o$), the star graph is a suppressor of fixation (*SoF*) whereas, under uniformly initialised Bd updating (Bd$^p$) the star is an amplifier of selection (*AoS*). **B** The uniformly initialised star graph under dB updating (dB$^o$) is a suppressor of selection (*SoS*). A previously unexplored category of graphs, amplifiers of fixation

(*AoF*), is introduced here. An amplifier of fixation has higher fixation probability for a mutant, regardless of its fitness, than on the complete graph. Under temperature initialised dB updating (dB$^p$) the star graph is an amplifier of fixation. Specifically, for finite $N$ the star graph is a piecewise amplifier of fixation, and only in the limit of infinite population size, it becomes an universal amplifier of fixation. The result that the star graph is an *AoF* for dB$^p$ updating is confirmed by simulations. Symbols correspond to dB simulations with 2000 independent runs for each graph (with each run conditioned on fixation). **C** Summary of how the choice of update rule affects the fixation dynamics for the star graph.

update rules, including some that have not been discussed here like bD and Db[37]. However, the star graph fails to amplify selection if the initial mutant is initialised according to the temperature initialisation scheme, where the central node is more likely to receive the initial mutant[38]. Under temperature initialised Bd updating, the star graph is instead a *SoF*[42], see Fig. 2A and a *SoS* for uniform initialised dB updating[47], see Fig. 2B.

So far, the star graph was not studied for the temperature initialised dB updating, because this requires the assumption that the parent moves instead of the offspring. Using the approach of recursive relations[37], in Supplementary Note 1 we derive the fixation probability of a mutant under dB updating on the star graph. When the mutant is initially placed on the centre node, its fixation probability is

$$\phi^{\bullet}_{\mathrm{dB},\star}(f',f) = \frac{N-1}{N\left(1 + \frac{N-2}{1+(N-1)\frac{f'}{f}}\right)}. \tag{5}$$

Here the star symbol denotes the star graph, and the filled circle in the superscript of $\phi^{\bullet}_{\star}$ indicates that the initial mutant is at the centre node. Similarly, the fixation probability for a mutant initially placed on a leaf node is

$$\phi^{\circ}_{\mathrm{dB},\star}(f',f) = \frac{\frac{f'}{f}}{\left(N-2+2\frac{f'}{f}\right)\left(1+\frac{N-2}{1+(N-1)\frac{f'}{f}}\right)}. \tag{6}$$

From Eqs. (5) and (6), we can compute the temperature initialised fixation probability of a mutant on the star graph under dB updating (equivalent to dB$^p$) as

$$\Phi^{\mathcal{T}}_{\mathrm{dB},\star}(f',f) = \frac{(N-1)\phi^{\bullet}_{\mathrm{dB},\star}(f',f) + \phi^{\circ}_{\mathrm{dB},\star}(f',f)}{N}. \tag{7}$$

From Eq. (7), we find that for dB$^p$ updating, the probability to fix advantageous mutants on the star graph is higher than in the well-mixed population, see Fig. 2B. But the fixation probability for deleterious mutants is also higher than the fixation probability on the complete graph, contrary to the original definition of *AoS*s where the probability to fix deleterious mutant is lower[13]. Therefore, it represents a new category of graphs which we call amplifier of fixation (*AoF*), where the probability to fix a mutant is higher than that of the complete graph, regardless of the mutant fitness value.

The star graph under temperature initialised dB updating dB$^p$ is a piecewise *AoF* for finite $N$, see Supplementary Fig. 1, and only in the limit $N \to \infty$, it is a universal *AoF*, see Fig. 2B. Intuitively, in a very large population, the initial mutant will most likely appear at the central node. Assuming the mutant does appear at the central node, in the next step an individual is selected with uniform probability to die. Most likely this is a leaf node. In this case, the mutant at the central node replaces the dead individual. This way, the initial mutant can survive in the population with higher probability, regardless of its fitness. Taking the $N \to \infty$ limit of the fixation probability profile we find,

$$\lim_{N\to\infty} \Phi^{\mathcal{T}}_{\mathrm{dB},\star}(f',f) = \lim_{N\to\infty} \phi^{\bullet}_{\mathrm{dB},\star}(f',f) = \frac{f'}{f+f'}, \text{ for } f' > 0. \tag{8}$$

Thus the probability to fix a deleterious mutant ($f' < f$) remains non-zero even in the limit of $N \to \infty$, which contradicts the conventional intuition that deleterious mutants are efficiently purged from large populations. A similar phenomenon was observed in refs. 9,52 in an explicitly spatial setup. The result of non-zero fixation probability for deleterious mutant remains robust even when parent movement is allowed with a small probability in the offspring movement dB process. For details, see Supplementary Note 2. The choice of individual type moving to vacant sites−parent or offspring−only affects the initial

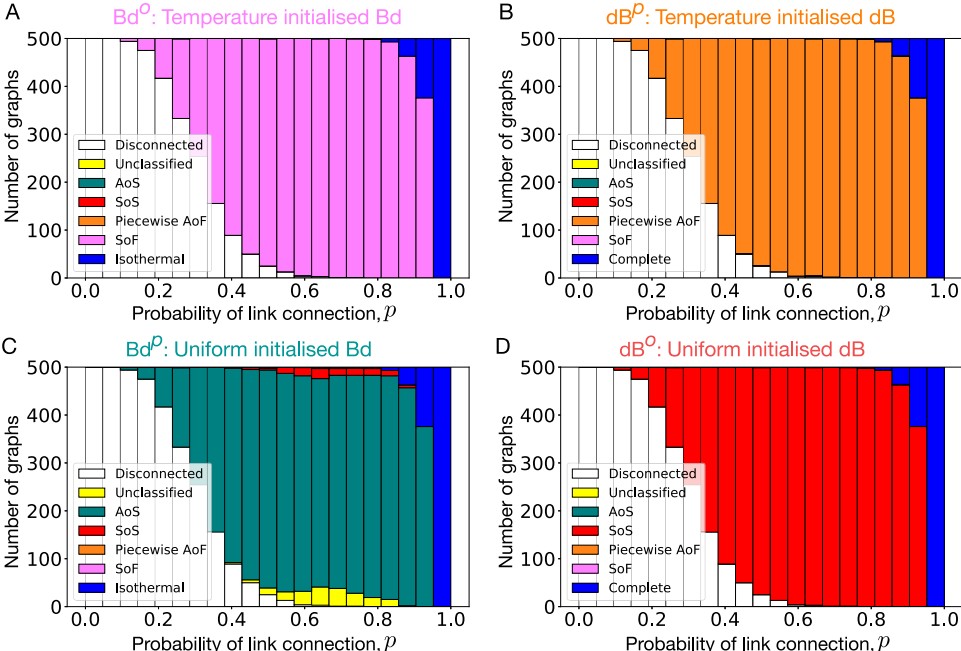

**Fig. 3 | Suppressors and amplifiers of fixation are ubiquitous and easy to construct.** We generated Erdős Rényi graphs of size $N = 8$ for several values of the probability of link connection, $p$. Fixation probability profiles for connected graphs are numerically obtained for temperature/uniform initialised Bd and dB updating. Refer to main text for details on the fitness discretisation. **A** Most of the graphs are suppressors of fixation under temperature initialised Bd updating, i.e., most of the graphs have lower fixation probability for a mutant regardless of its fitness than the complete graph, whereas, (**B**) most of the connected random graphs are piecewise amplifiers of fixation under temperature initialised dB updating. These graphs have higher fixation probability than the complete graph for mutants with fitness $f' \leq f^*$, and lower fixation probability for $f' > f^*$ with $f \geq 1$. In other words, within our chosen resolution of the fitness scale, the fixation of deleterious mutations is amplified and the fixation of beneficial mutations is suppressed. Similarly, (**C**) most of the connected graphs are amplifiers of selection under uniformly initialised Bd updating, whereas, (**D**) most of the connected random graphs are suppressor of fixation under uniformly initialised dB updating. Here (**C**, **D**) is used only for comparison, obtained by a similar analysis performed in ref. 33. In Supplementary Fig. 3, the fixation probabilities of random graphs with $p = 0.5$ are shown for different update schemes.

mutant placement on the structure. In the absence of mutations, these two choices are equivalent as both the offspring resemble the parent type. Therefore, for the process where both offspring and parent can move, the corresponding fixation probability is the convex combination of uniform and temperature initialised fixation probabilities.

**Numerical classification of graphs.** In the previous section, we have analysed the star graph under 4 different updating schemes, $Bd^o$, $dB^o$, $Bd^p$, and $dB^p$. Depending in the update scheme, the star graph can be a *SoF*, a *SoS*, an *AoS*, or an *AoF*. The classification carries over to other graphs and we implement it as follows: we numerically compute the fixation probability for a mutant with fitness values $f' = 0.5, 0.75, 1, 1.25, 1.5, 1.75, 2, 2.25$ and $2.5$, see Supplementary Note 3 for more details. With wild-type individual fitness $f = 1$, we classify a given connected graph $G$ as

- *AoS*, if $\Phi_G^{\mathcal{T}}(f',f) < \Phi_C(f',f)$ for $f' = 0.5, 0.75$ and $\Phi_G^{\mathcal{T}}(f',f) > \Phi_C(f',f)$ for $f' \geq 1.25$.
- *SoS*, if $\Phi_G^{\mathcal{T}}(f',f) > \Phi_C(f',f)$ for $f' = 0.5, 0.75$ and $\Phi_G^{\mathcal{T}}(f',f) < \Phi_C(f',f)$ for $f' \geq 1.25$.
- *SoF*, if $\Phi_G^{\mathcal{T}}(f',f) < \Phi_C(f',f)$ for all $f'$.
- *AoF*, if $\Phi_G^{\mathcal{T}}(f',f) > \Phi_C(f',f)$ for all $f'$.
- *Piecewise AoF*, if $\Phi_G^{\mathcal{T}}(f',f) > \Phi_C(f',f)$ for $f' \leq f^*$ and $\Phi_G^{\mathcal{T}}(f',f) < \Phi_C(f',f)$ for $f' > f^*$, where $f \geq 1$.
- *Isothermal* graph if $\Phi_G^{\mathcal{T}}(f',f) = \Phi_C(f',f)$ for all pairs of $f',f$, and every node has the same degree. However, the isothermal graphs have the same fixation probabilities as the complete graph only for Bd updating[35] (see Supplementary Note 4 for further discussion).

Note that these definitions assume that at neutrality ($f' = f$) all the graphs have the fixation probability $1/N$, as the complete graph which

is chosen as the basis for comparison: For an arbitrary graph, every node when initialised with a mutant can have different fixation probability. The sum of those probabilities is equal to 1 as one of the individuals will eventually fix. Therefore, the uniform initialised fixation probability on any graph is $1/N$. However, this is true only under uniform mutant initialisation. For an arbitrary graph with temperature initialisation, the average of individual node fixation probabilities has to be weighted thus and it does not have to equal $1/N$[53]. We note that piecewise amplifiers and suppressors are sometimes referred to as transient in the literature[47,54].

**Random graphs.** Based on this, we study the fixation probability profiles for random graphs of size 8 for the same update rules, to see to what extent the observations made for the star graph extend to random graphs. For this purpose, we randomly generated Erdős Rényi graphs[55] for different probabilities of link connection, $p$. Setting $p = 0$ generates fully disconnected graphs whereas $p = 1$ generates the complete graph. As the fixation probability is defined only if the graph is connected, we condition on connected graphs[56,57]. From Fig. 3A, we find that, just as the star graph, most random graphs are *SoF* under the temperature initialised Bd process (equivalent to $Bd^o$). Similarly, most of the random graphs under temperature initialised dB (equivalent to $dB^p$) are (piecewise) *AoF*, see Fig. 3B. Therefore, *AoF* and *SoF* are ubiquitous. Under the uniformly initialised Bd process (equivalent to $Bd^p$) and dB process (equivalent to $dB^o$), most of the random graphs are *AoS* and *SoS* respectively, see Fig. 3C, D. The ubiquity of these categories has been shown earlier in ref. 33.

**Long time scales: mutation-selection balance**
After studying short-term fixation dynamics in graph-structured populations, we now move our focus to long-term mutation-selection

dynamics. We assume that the state space is a bounded fitness interval $[f_{min}, f_{max}]$. During a birth event, with probability $1 - \mu$ the offspring resembles its parent and has the same fitness, and otherwise it mutates to a new fitness sampled from a mutational fitness distribution $\rho(f', f)$[58]. Here $\rho(f', f)$ denotes the probability density of the mutant offspring fitness $f'$ given the parental fitness $f$. Assigning the fitness of new mutations at random from a continuous distribution corresponds to an infinite allele model, where each mutation leads to a new type and mutations cannot be reverted. When $\rho(f', f)$ is independent of the initial fitness $f$ this corresponds to Kingman's House-of-Cards (HoC) model[59,60].

We work in the regime of low mutation rates, $\mu \ll 1$ where the population is monomorphic almost all the time, except during a fixation event. All individuals have the same fitness value, which can be used to label the entire population. Specifically, the average time between two successive mutations is large enough so that the initial mutant reaches fixation or goes extinct before the next mutation appears[61]. Any change in the state of the population requires the fixation of a new mutation. Thus, the fixation probability and the mutant initialisation (depending on the details of the update rule) $\Phi_G^{\mathcal{T}}$ fully determine the long-term mutation-selection dynamics. These dynamics are known by multiple names, e.g., sequential dynamics, periodic selection[25,62], or origin-fixation dynamics[63].

The origin-fixation dynamics on a population structure $G$ is a continuous time Markov chain on the fitness state space governed by the master equation

$$\frac{\partial P_G(f, t)}{\partial t} = \int df' \underbrace{\Phi_G^{\mathcal{T}}(f, f')\rho(f, f')\mu}_{T_{f \leftarrow f'}} P_G(f', t) - \int df' \underbrace{\Phi_G^{\mathcal{T}}(f', f)\rho(f', f)\mu}_{T_{f' \leftarrow f}} P_G(f, t), \tag{9}$$

where $P_G(f, t)$ is the probability density function for the structure $G$ to be in between fitness state $f$ and $f + df$ at time $t$.

At long times a steady-state fitness distribution $P_G^*(f)$ is attained. $P_G^*(f)$ satisfies the stationarity condition

$$\int df' \, T_{f \leftarrow f'} P_G^*(f') = \int df' \, T_{f' \leftarrow f} P_G^*(f), \tag{10}$$

where $T_{f \leftarrow f'}$ is the transition probability from the fitness state $f'$ to $f$ and $T_{f' \leftarrow f}$ is the transition probability from the fitness state $f$ to $f'$ as defined in the Eq. (9). This condition simplifies considerably if the Markov chain is reversible, which implies the detailed balance relation[64,65]

$$T_{f \leftarrow f'} P_G^*(f') = T_{f' \leftarrow f} P_G^*(f), \quad \text{for all } f', f. \tag{11}$$

Normalising $P_G^*(f')$, the steady-state solution takes the form,

$$P_G^*(f) = \frac{1}{\int df' \frac{T_{f' \leftarrow f}}{T_{f \leftarrow f'}}} = \frac{1}{\int df' \frac{\Phi_G^{\mathcal{T}}(f', f)}{\Phi_G^{\mathcal{T}}(f, f')} \cdot \frac{\rho(f', f)}{\rho(f, f')}}$$
$$= \frac{1}{\int df' \, \Psi_G^{\mathcal{T}}(f', f) \cdot \frac{\rho(f', f)}{\rho(f, f')}}. \tag{12}$$

Here, we have introduced the ratio of fixation probabilities $\Psi_G^{\mathcal{T}}(f', f)$. In[66] it has been shown that the origin-fixation dynamics is reversible if and only if $\Psi_G^{\mathcal{T}}(f', f)$ is a power-law, i.e.,

$$\Psi_G^{\mathcal{T}}(f', f) = \left(\frac{f'}{f}\right)^{\nu}, \tag{13}$$

where $\nu$ is constant. For most graphs, this condition is not satisfied exactly, but it may hold approximately in a range of fitness values and/or for large population sizes, see Supplementary Note 5 for more details. Eq. (12) can then be used as an approximation to the steady state fitness distribution[42].

Expression (12) can be simplified further in the HoC setting, where $\rho(f', f) = \rho(f')$. Using (13) one obtains

$$P_G^*(f) = \frac{1}{M^{(\nu)}} f^{\nu} \rho(f), \tag{14}$$

where $M^{(\nu)} = \int df' \, (f')^{\nu} \rho(f')$ is the $\nu$th moment of $\rho$. This proves in particular that a stationary distribution exists whenever all moments of the mutant fitness distribution $\rho(f)$ are finite. The factor $f^{\nu}$ accounts for the bias of the distribution towards higher fitness values introduced by selection. In the following, we adopt the HoC setting and choose the mutant fitness distribution $\rho(f)$ to be uniform on the interval $[f_{min}, f_{max}]$. The generalization to other mutant fitness distributions is straightforward using (14). From now onwards, we use the notation $P_G$ for $P_G^*$.

**Complete and regular graphs.** For the complete graph under dB updating, $\Psi_G^{\mathcal{T}}(f', f)$ takes the form

$$\Psi_C^{\mathcal{T}}(f', f) = \Psi_C(f', f) = \frac{\Phi_C(f', f)}{\Phi_C(f, f')} = \left(\frac{f'}{f}\right)^{N-2}. \tag{15}$$

Thus, the Moran dB origin-fixation dynamics on the complete graph is reversible with $\nu = N - 2$. Similarly, for the Moran Bd origin-fixation dynamics, in[66,67] reversibility is shown to hold for the complete graph with $\nu = N - 1$. Using (14), the steady-state fitness distribution for dB dynamics is thus given by

$$P_C(f) = \frac{N - 1}{f_{max}^{N-1} - f_{min}^{N-1}} f^{N-2} = \frac{N - 1}{f_{max}^{N-1} - f_{min}^{N-1}} e^{(N-2)\log f}, \tag{16}$$

and the steady-state average fitness is

$$\langle f \rangle_C = \int df \, f P_C(f) = \frac{N - 1}{N} \frac{f_{max}^N - f_{min}^N}{f_{max}^{N-1} - f_{min}^{N-1}}. \tag{17}$$

In the limit $N \to \infty$,

$$\lim_{N \to \infty} \Phi_C(f', f) = \begin{cases} 1 - \frac{f}{f'}, & \text{if } f' > f, \\ 0 & \text{otherwise}. \end{cases} \tag{18}$$

Thus, under this dynamics, an infinitely sized well-mixed population can only move forward on the fitness space with long-term fitness converging to $f_{max}$. This can also be seen by performing the limit of $N \to \infty$ on the average steady-state fitness in Eq. (17). The same result holds for the Moran origin-fixation Bd dynamics.

The steady-state fitness distribution in Eq. (16) takes the form of an exponential, Boltzmann-like distribution for $-\log f$. This suggests an analogy between statistical mechanics and evolutionary theory that has been pointed out in numerous former studies[67–71]. Under this analogy, a physical system's energy is equivalent to the negative logarithm of fitness, and the inverse physical temperature is equivalent to the effective population size. Just like high physical temperatures result in strong thermal fluctuations, low effective population sizes lead to highly stochastic population dynamics.

For general (reversible) origin-fixation dynamics, the role of $N - 2$ or $N - 1$ for the Moran model on the complete graph is taken over by the exponent $\nu$ in Eq. (13). This suggests to define $\nu$ as a measure of the effective population size of a general graph (see[12,72,73] for definitions of effective population sizes in other contexts). While this definition is

strictly applicable only if the long-term origin fixation dynamics is reversible, it is useful also when reversibility holds approximately or asymptotically. We will see below that the definition is consistent with intuition, in the sense that graphs with larger (smaller) effective population sizes display smaller (larger) fluctuations in the evolutionary dynamics. We also studied the long-term dB mutation-selection dynamics for other regular graphs, see Supplementary Note 4 for details. The one-dimensional cycle graph, a *SoF* under dB updating[33,54], has a lower probability of fixing mutants regardless of the fitness of the mutant compared to the well-mixed population. The cycle graph is worse at fixing beneficial mutations but it is better at preventing the fixation of deleterious mutants. The ratio of the fixation probabilities $\Psi_\circ$ is exactly the same as that for the complete graph, i.e.,

$$\Psi_\circ(f', f) = \Psi_C(f', f) = \left(\frac{f'}{\bar{f}}\right)^{N-2}, \tag{19}$$

see Eq. 24 in the Supplementary Information text. In the long run, the lower probability of fixing beneficial mutants for the cycle graph gets compensated by the higher probability of rejecting deleterious mutants. As a result, under Moran dB origin-fixation dynamics, the steady-state statistics for the cycle graph are the same as for the complete graph. Similarly, the two-dimensional grid with periodic boundary conditions, although having a different fixation probability profile[35], approximately attains the same steady-state statistics for the long-term mutation-selection dynamics as the complete graph, see

Supplementary Note 4 for more details. Therefore, for long-term dB dynamics, we hypothesize other (large) regular graphs to have the same steady state as the complete graph. For Moran Bd updating, the complete graph, the cycle graph, and the two-dimensional lattice (with periodic boundary conditions) have the same fixation probabilities[13], therefore they all have the same steady-state statistics for the Moran Bd origin-fixation dynamics.

**Star graph**. We now study long-term mutation-selection dynamics on the star graph under Moran Bd$^p$, Bd$^o$, dB$^p$, and dB$^o$ updating. In the long-term Moran dB$^o$ origin-fixation dynamics, the complete graph leads to a higher average fitness than the star graph, see Fig. 4B. This is expected, as the star graph under dB$^o$ updating is a *SoS*.

For the Moran dB$^p$ origin-fixation dynamics, the star has a higher probability to fix mutants than the complete graph. However, in the long-term dynamics, the complete graph leads to a higher average fitness than the star graph. Moreover, the star graph with dB$^o$ updating attains higher fitness than the star graph subjected to dB$^p$ updating. This happens because the star graph under dB$^p$ updating has a higher probability to fix deleterious mutants. As the population gets closer to the fitness peak (here $f_{max}$), the probability for the mutations to have deleterious fitness effects also increases. Consequently, the fate of deleterious mutants has a strong influence on the steady-state fitness of the population.

To understand this further, we study the large $N$ behavior of the steady-state for the various Moran origin-fixation dynamics on the star

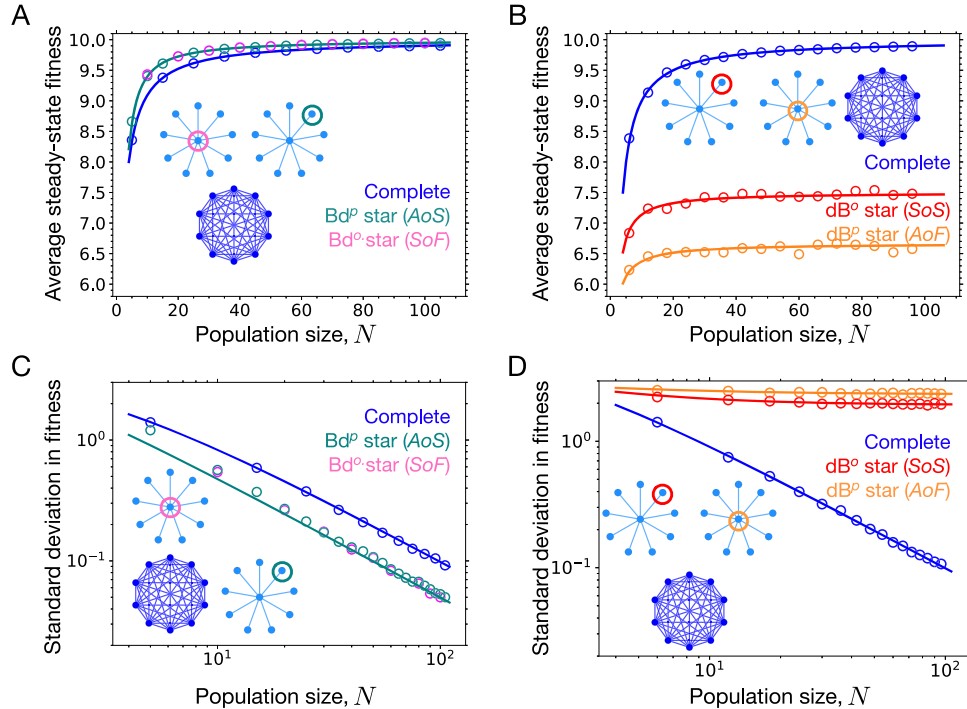

**Fig. 4 | Moran origin-fixation dynamics on the star graph.** The figure shows the average and standard deviation of the steady-state fitness distribution as a function of population size. Here, the mutational fitness distribution is uniform. Circles represent results obtained from Monte Carlo simulations of the origin-fixation Markov chain, and lines were computed from the approximate expression Eq. (12) for the steady state fitness distribution. **A** For Moran Bd$^o$ origin-fixation dynamics, the star graph, despite being a suppressor of fixation, attains not only higher average fitness than the complete graph but identical fitness as the star graph under Bd$^p$ dynamics, where it is an amplifier of selection. This happens because the Bd$^o$ star compensates for its inability to fix beneficial mutants by rejecting deleterious mutants efficiently. **B** For dB$^p$ dynamics, the star graph, being an amplifier of fixation, not only attains lower steady-state average fitness than the well-mixed population but also lower than the star graph

subjected to dB$^o$ update, where it is a suppressor of selection. This happens because the dB$^p$ star is worse in rejecting deleterious mutations than the dB$^o$ star. Therefore, being good at fixing beneficial mutants is not sufficient to attain higher fitness in the long-term evolution. **C** The star graph under Bd long-term dynamics not only attains higher average fitness but also lower fluctuations in the steady-state than the well-mixed population. This can be understood by the higher effective population size of the star graph. **D** Compared to the average fitness order under dB dynamics in panel B, the order for the standard deviation in fitness is reversed: dB$^p$-star, dB$^o$-star, and the complete graph. Moreover, the standard deviation for the star graphs under dB long-term dynamics saturates to finite values for large $N$, as their effective population sizes are independent of $N$. Parameters: $f_{min} = 1$, $f_{max} = 10$ and the number of independent runs for Monte Carlo simulations is 2000.

graph. In the limit of large $N$ for Moran $dB^p$ updating, we find from Eq. (8) that

$$\Psi_{dB^p,\star}(f',f) \approx \frac{f'}{f}. \tag{20}$$

From Eq. (20), we find that the effective population size in the large $N$ limit for the star graph under $dB^p$ updating is $\nu = 1$. The corresponding steady-state fitness distribution in the large $N$ limit is

$$P_{dB^p,\star}(f) = \frac{2f}{f_{max}^2 - f_{min}^2} \tag{21}$$

and the steady-state average fitness is

$$\langle f \rangle_{dB^p,\star} = \int df\, f P_{dB^p,\star}(f) = \frac{2}{3}\frac{f_{max}^3 - f_{min}^3}{f_{max}^2 - f_{min}^2} \approx \frac{2}{3}f_{max} \tag{22}$$

for $f_{max} \gg f_{min}$. Similarly, the large $N$ limit for the star graph under $dB^o$ updating gives (see Supplementary Note 1 for details)

$$\Phi_{dB^o,\star}(f',f) \approx \frac{1}{N}\frac{f'}{f} \tag{23}$$

and therefore

$$\Psi_{dB^o,\star}(f',f) \approx \left(\frac{f'}{f}\right)^2,$$
$$P_{dB^o,\star}(f) = \frac{3f^2}{f_{max}^3 - f_{min}^3}. \tag{24}$$

In the limit of large $N$, the effective population size of the star graph subjected to $dB^o$ dynamics is twice that obtained for $dB^p$ dynamics. As a consequence, the $dB^o$ star graph attains higher average fitness in the steady-state,

$$\langle f \rangle_{dB^o,\star} = \int df\, f P_{dB^o,\star}(f) = \frac{3}{4}\frac{f_{max}^4 - f_{min}^4}{f_{max}^3 - f_{min}^3} \approx \frac{3}{4}f_{max} \tag{25}$$

for $f_{max} \gg f_{min}$. Although $\lim_{N\to\infty} \Phi_{dB^o,\star}(f',f) = 0$ and $\lim_{N\to\infty} \Phi_{dB^p,\star}(f',f) \neq 0$, we find $\langle f \rangle_{dB^o,\star} > \langle f \rangle_{dB^p,\star}$ (but note that, because of the lower fixation probability, the higher steady-state fitness of the $dB^o$ star is attained after a longer time). The effective population size and hence the steady-state fitness statistics for the process where parent individuals are allowed to move with a non-zero probability is the same as for the case where only parents move. This shows that a very small proportion of parent moving for a predominantly offspring moving dB process is sufficient to significantly affect the long-term evolution. See Supplementary Note 2 for more details.

Performing a similar analysis for Birth-death updating, we find that the star graph under $Bd^p$ and $Bd^o$ updating satisfies

$$\Psi_{Bd^p,\star}(f',f) \approx \left(\frac{f'}{f}\right)^{2N-2} \approx \Psi_{Bd^o,\star}(f',f). \tag{26}$$

The first approximation follows from ref. 50 and the second approximation follows from ref. 42. What this means is that, although under Moran $Bd^p$ and $Bd^o$ updating the star has quite different fixation probability profiles, see Fig. 2A, in the long term to a good approximation they display identical steady-state statistics because in both cases the star graph has the same effective population size of $2N$. This also means that the star graph under long-term Moran Bd dynamics attains higher average fitness in the steady state than the well-mixed

population. Specifically, the steady-state fitness distribution is

$$P_{Bd^p,\star}(f) = P_{Bd^o,\star}(f) = \frac{2N-1}{f_{max}^{2N-1} - f_{min}^{2N-1}}f^{2N-2}$$
$$= \frac{2N-1}{f_{max}^{2N-1} - f_{min}^{2N-1}}e^{(2N-2)\log f} \tag{27}$$

and the average steady-state fitness is

$$\langle f \rangle_{Bd^p,\star} = \langle f \rangle_{Bd^o,\star} = \frac{2N-1}{2N}\frac{f_{max}^{2N} - f_{min}^{2N}}{f_{max}^{2N-1} - f_{min}^{2N-1}} \tag{28}$$

with

$$\langle f \rangle_{Bd,\star} - \langle f \rangle_C \approx \frac{f_{max}}{2N}. \tag{29}$$

The star graph under $Bd^p$ updating, an *AoS* (Fig. 2A), expectedly attains higher fitness than the well-mixed population because it is better in fixing beneficial mutations and preventing the fixation of deleterious mutations. However, the star graph under $Bd^o$ updating, a *SoF*, also attains a higher steady-state average fitness than the complete graph (identical to the one under $Bd^p$ updating) because it is much better at rejecting deleterious mutants, which compensates for its lower probability to fix beneficial mutations[42].

The effective population size also affects the fluctuations in the steady state. Because the effective population size of the star graph under dB updating is independent of $N$, the standard deviation in fitness does not change at large $N$. The *AoF* star experiences higher fluctuations than the *SoF* star because of its lower effective population size, see Fig. 4D. Under Bd updating, the effective population size of the star graph is twice the effective population size of the complete graph. Therefore, the star experiences lower fluctuations than the complete graph, and the standard deviation decreases with increasing $N$, see Fig. 4C. For more details, see Supplementary Note 6.

**Random graphs.** For long-term evolution on the star graph, the deleterious mutant regime can substantially affect the fate of the dynamics. Does this effect extend to other graphs? Can we expect the *SoF* that we found in Fig. 3A to have higher long-term fitness than the well-mixed population? The answer in not obvious. Similarly, what can we say about the long-term fitness fate of the *AoF* found in Fig. 3B? Do all of them attain lower steady-state average fitness than the complete graph, just like the star graph under $dB^p$ updating? We explore these questions next. We move forward by discretising the fitness space and study the Moran $Bd^p$, $Bd^o$, $dB^p$, and $dB^o$ origin-fixation dynamics for several random graphs. Steady-state statistics of these graphs are obtained by solving the respective Markov chains numerically, see Supplementary Note 7 for details.

Computing the steady-state average fitness for all connected random graphs, in Fig. 5A we find that for long-term $Bd^o$ dynamics, almost all *SoF* attain higher steady-state average fitness than the complete graph, whereas in Fig. 5B for long-term $dB^p$ dynamics, all the piecewise *AoF* attain lower steady-state average fitness. Interestingly, for the case of $Bd^p$ dynamics where most of the connected graphs are *AoS*, the graphs attain quite similar average fitness as they do when subjected to the $Bd^o$ dynamics, see Fig. 5C. In Fig. 4A, we have seen that the star graph attains the same fitness for the two kinds of Bd long-term dynamics. Now we confirm this for all other graphs. Expectedly, for the long-term $dB^o$ dynamics where most of the random connected graphs are *SoS*, the majority of random graphs attain lower average fitness, see Fig. 5D.

It is difficult to find general conditions under which a structured population has higher average steady-state fitness than the complete graph. In Supplementary Note 8, we derive a sufficient condition for

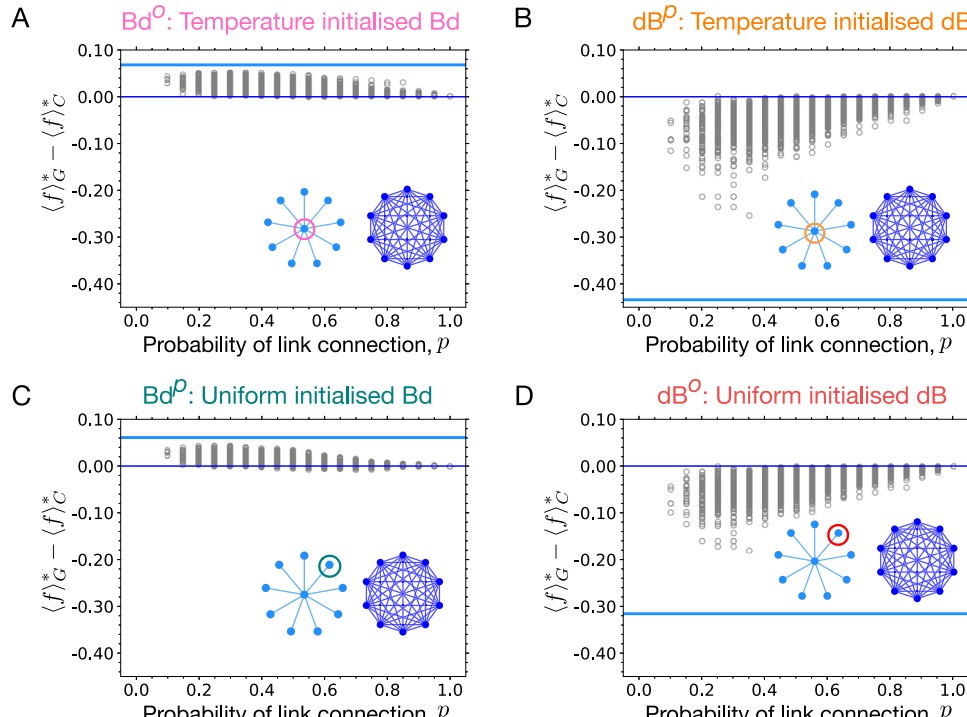

**Fig. 5 | Suppressors of fixation attain higher long-term fitness whereas amplifiers of fixation attain lower fitness.** Numerical solutions of the Markov chain on the fitness space for several Erdős-Rényi graphs of size 8 are presented. The fitness space is discretised into fitness values $f_0 = 0.5, f_1 = 0.75, f_2 = 1, f_3 = 1.25, f_4 = 1.5, f_5 = 1.75,$ and $f_6 = 2$. For new mutations, nearest neighbour jumps in the discretised fitness space are considered. The bold horizontal lines represent the steady-state fitness obtained on the star graph relative to the complete graph (thin horizontal line) under different dynamics. **A** Most of the random graphs are suppressors of fixation under Bd$^o$ updating, yet they attain higher fitness than the well-mixed population. **B** Similarly, most of the random graphs are (piecewise) amplifiers of fixation under dB$^p$ updating, yet they attain lower steady-state average fitness. **C, D** Expectedly, amplifiers of selection attain higher fitness for Bd$^p$ dynamics, and suppressors of selection attain lower fitness for dB$^o$ dynamics. Thus, the deleterious mutant regime is important for generic graph structures when subjected to long-term dynamics.

this. We use this condition to determine the ordering of the average steady-state fitnesses for the structures reported in Fig. 4B and show that they are consistent with the numerical results.

## Discussion

Most of the initial research in evolutionary graph theory has focused on the Moran Birth-death (Bd) update with uniform initialisation[13,26,50]. The uniform initialisation is typically justified by considering spontaneous mutations during an individual's lifetime[74]. However, when mutations occur during reproduction instead[38], justifying the use of uniform initialised Bd updating becomes more challenging. In that scenario, temperature initialisation is a more natural choice. Our findings demonstrate that the uniform initialisation in the Bd update naturally arises when parent-type offspring move to vacant nodes (Bd$^p$) rather than mutant offspring individuals (Bd$^o$). Furthermore, by necessitating parent-type individuals to move instead of mutant offspring individuals, we have uncovered the existence of temperature-initialised dB updating (dB$^p$), an update scheme previously considered non-existent. In conclusion, we emphasize that a mutant initialisation scheme is an outcome of an update rule and the mode of mutations, and need not be specified on top of it. An update rule should be sufficient to generate the fixation dynamics on a graph.

Moran dB$^p$ updating introduces a new category of graphs known as *AoF* (amplifiers of fixation), where the fixation probability is higher regardless of the fitness values compared to the complete graph. The star graph under dB$^p$ updating is an *AoF* with non-zero probability to fix deleterious mutants, even in the limit of infinite population size. For all other previously known graphs, such as *AoS, SoS, SoF*, the complete graph, the probability of fixing any deleterious mutant goes to zero for

large population sizes. Consequently, the deleterious mutant regime has not been extensively explored in the literature. The discovery of *AoF* underscores the need to consider the deleterious mutant regime in graph classification, and its importance in the computation of fixation probability and time.

The results derived from the analysis of the star graph for different update rules also extend to Erdős-Rényi random graphs. Specifically, we observe that the majority of small random graphs are *SoF* under Bd$^o$ updating and piecewise *AoF* under dB$^p$ updating. This finding closely resembles the result of ref. 33, where most random graphs were identified as *AoS* under Bd$^p$ updating and *SoS* under dB$^o$ updating. Consequently, it is not only the order of birth and death events but also the choice of the individual moving to vacant sites that significantly impacts the results at the short-term fixation time scale. Earlier work in evolutionary graph theory has focused on designing strong amplifiers of selection—structures with a high probability of fixing beneficial mutants[13,29,75,76]. Our work offers new research directions where structures can be designed to obtain desirable fixation profiles, both for the beneficial and deleterious mutant regimes. Different update rules allow to manipulate the fixation probabilities of beneficial and deleterious mutations independently.

The choice of moving individuals also affects the long-term Moran origin-fixation dynamics. The star graph, an *SoF* under Bd$^o$ updating, despite having lower probability to fix advantageous mutants attains higher fitness in the long-term dynamics than the well-mixed population. Similarly, under dB$^p$ updating, the star graph despite being an *AoF* with higher probability of fixing beneficial mutants attains lower fitness. In the former case, the star graph is better in rejecting deleterious mutants, compensating for its lower probability to fix beneficial mutations. In the latter scenario, the star

graph is not good is preventing the fixation of disadvantageous mutants despite being better at fixing beneficial mutations.

More concretely, the effective population sizes of the star graph for different updating explain the corresponding steady-states and the contribution coming from the deleterious mutant regime. The *SoF* star graph has a higher effective population size than the *AoF* star graph. Additionally, the results obtained for the long-term evolution on the star graph also extend to random graphs. Under Bd$^o$ updating, most of the random graphs, despite being *SoF*, attain higher fitness than the isothermal graphs. Whereas, under dB$^p$ updating, most of the random graphs despite being piecewise *AoF* attain lower fitness than the isothermal graphs.

To summarise, care should be taken before speculating about the fate of long-term evolution on spatial structures based on the short-term fixation dynamics. For a population adapting on a fitness landscape, the outcome depends on two factors. First, how effective the population is in stepping forward, and second, how good it is in not falling backward. The effect of deleterious mutant regime can also be seen in the transient phase of evolutionary dynamics[58]. The likelihood for the average fitness trajectory to be non-monotonic increases with the probability to accept deleterious mutations, see Supplementary Note 9 for more details. Overall, the deleterious mutant regime is important when it comes to long-term evolution on spatial structures, something that is often ignored in adaptive evolution theories with well-mixed populations. As the present work has focused on long-term evolution in the regime of low mutation rates and origin-fixation dynamics, the role of deleterious mutations for structured populations subject to a large supply of mutations should be addressed in future research. One such direction is to understand valley crossing on rugged fitness landscapes[46,77].

Experiments with microbial populations have begun to systematically compare evolution in well-mixed and structured environments[10,11,78]. For example, the weakened selection against deleterious mutations in spatial environments predicted by theory has been confirmed experimentally for yeast cells that undergo an irreversible conversion to a low-fitness type[8]. Another recent study demonstrates a case of suppression of selection and genetic drift in *E.coli* biofilms on corrugated surfaces[79]. Moreover, evolution experiments designed to test specific predictions of evolutionary graph theory have been performed with *Pseudomonas aeruginosa*[80], and it is only a matter of time before further studies are conducted along these lines[81]. The quantitative description of such experiments necessitates the extension of evolutionary graph theory to the structured metapopulation level[34,82–86]. While our work focuses on the setting of one-node-one-individual, extending to network-structured metapopulations is an important future direction. For some update rules, it is known that the results at fixation time scale carry over to the metapopulation case[84], but this is a largely unexplored area[34]. Indeed, the analysis in this paper can be adapted for the study of metapopulations, and it is found that, provided the central deme is initialized with mutants, the metastar structure presented in ref. 85 behaves like an amplifier of fixation or a suppressor of fixation depending on the migration asymmetry between demes, see Supplementary Note 10 for more details. In this way, the results presented in this paper can gain empirical relevance[81]. There are experimental challenges associated with the separation of the time-scales of migration and fixation in a deme as assumed here, and the metastar is predicted to behave differently when this assumption is relaxed[87]. Nonetheless, our work indicates that examining the role of the deleterious mutant regime for structured metapopulations is an important future direction.

## Methods
The fixation probabilities for the star graph and the Erdős-Rényi are presented in the Results, with derivations/details provided in the Supplementary Information file under headings, Exact formula for the fixation probability of a mutant on the star graph under dB and Bd updating, and Matrix approach to compute fixation probability on a random graph.

The derivations for the results on long-term evolution across various graphs are covered in the Supplementary Notes 4 through 7 under headings, Long-term evolution on regular graphs, Reversibility, Standard deviation in the steady-state fitness distribution, and Long-term evolution on discrete fitness space.

## Data availability
The data to reproduce our figures is available on Zenodo[88], https://zenodo.org/records/14259904.

## Code availability
The source code (Mathematica files / Jupyter notebooks) to reproduce our figures is available on Zenodo[88], https://zenodo.org/records/14259904.

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

## Acknowledgements
We thank Alex McAvoy, Helen Alexander, Yuriy Pichugin, Joshua Plotkin, and Arjan De Visser for helpful discussions. We thank Carsten Fortmann-Grote for computational support. This work was supported by the Max Planck Society and by DFG within CRC 1310 'Predictability in Evolution'.

## Author contributions
N.S., S.G.D., J.K., and A.T. designed research; performed research; analyzed data; and wrote the paper.

## Funding

## Competing interests
The authors declare no competing interest.
