## [Transparent Peer Review file · Nature Communications]

Graph-structured populations elucidate the role of deleterious mutations in long-term evolution

Corresponding Author: Dr Nikhil Sharma

Version 0:

Reviewer comments:

Reviewer #1

(Remarks to the Author)

In the paper "Graph-structured populations elucidate the role of deleterious mutations in long-term evolution" by N.Sharma et al, the authors investigate both short-term and long term evolutionary dynamics on graphs. To this end, they provide a very nice classification of updating and initialization schemes that appear in the Moran process on graphs, and show how different rules lead to very different properties of the system (suppressors/amplifiers of selection, and suppressors/amplifiers of fixation).

As someone working in this area I found this type of classification very useful, and the results fascinating. I have several comments that may help improve clarity of the paper and highlight its impact. These are in no particular order.

-- In the abstract, the authors use the terminology such as death-Birth and Birth-death. I think most people won't know what capitalization stands for. So maybe it is possible to explain it there. And, instead of listing all the results, maybe it is even more important to highlight the important points that are often overlooked (such as that it matters whether birth or death is under selection, and it matters where the mutant is placed).

-- Again, in the abstract, I suggest that more precise language is used: instead of "most random graphs" perhaps "most Erdos-Renyi graphs under 9 nodes".

-- The authors only considered the dB and Bd processes. Can they say something about Db and bD processes? Perhaps there is some symmetry in the patterns.

-- On page 4, the derivations correspond to directed graphs, but since the paper only deals with undirected graphs, this can be significantly shortened (to reduce confusion).

-- Type: l.105: rules rules (repetition)

-- The authors only consider the regime of sequential fixation. Would it be interesting to include somewhat higher mutation rates and see if the results change under stochastic tunneling?

-- Fig.2 (and other similar graphics): the graphics showing star graphs with the central node and one of the leaf nodes circled are confusing. As far as I understand, the authors are not comparing the probability of fixation of mutants that are put in the center vs the periphery. Instead, they are comparing the initial placement schemes denoted by "o" and "p", which result in a preferential placement in the center or in the periphery. Maybe this could be explained in the figure caption.

--In the Discussion section, some experiments are cited, for which these theories could be relevant. It would be great to give some more details of what biological systems are those?

(Remarks on code availability)

Reviewer #2

(Remarks to the Author)

This manuscript presents a study of spatially structured populations on graphs, in the framework of evolutionary graph theory. It sheds light on new possible impacts of graph structures under a new update rule. In particular, this update rule gives rise to suppressors of fixation (SoF), which decrease the fixation probability of all mutants compared to a well-mixed population. Interestingly, populations on these graphs can attain higher fitness in the long-term.

This manuscript is extremely complete, as it considers various aspects of a new update rule, including its relationship to the mutant initialization, its impact on fixation probability in simple graphs, but also in random graphs, and a detailed analysis of the impact on long-term fitness. I particularly appreciated the discussion of the relationship between update rules and initialization schemes under the assumption that mutations occur at division.

It is also scientifically sound and very well-presented, and overall clear and easy to read.

In my opinion, it constitutes an important advance in evolutionary graph theory.

However, I found some issues regarding the link to natural biological situations, and to other models and to experiments. They are spelled out in my major points below.

Other important points regard some clarifications and links to published works. They are detailed in my other points below. Finally, there are a few minor points and typos that I recommend correcting.

Major points:

1- While this manuscript presents a theoretical study, connecting with natural biological situations is important. At the moment, there are some points that are unclear or potentially misleading about this. I recommend clarifying them. In particular:

1a- Line 100: "lizards and snakes": it is not clear at all that the model presented here would be valid for them. First, these organisms are diploids that reproduce sexually, which is not modelled here. Second, they move actively in continuous space, meaning that a static graph may not be an adequate representation of spatial structure in their case. It would be good to either clarify and mitigate this, or to remove this example.

1b- Line 105: "microbial and somatic cell populations, where parent and offspring cannot necessarily be distinguished (...)

111 cell identical to the mother cell cannot move instead": These are very good points. But don't they argue for studying update rules where either the parent or the offspring may move, e.g. with 50% probability, rather than analysing the two extreme cases where only one of them moves? It would be good to discuss this.

1c- Line 219: "we assume that the mutant's fitness is sampled from a uniform distribution": It would be valuable to say a bit more about the realism of this assumption.

2- Similarly, the link with other models and with experiments can be improved, in particular:

2a- Line 374: "While our work focuses on the setting of one-node-one-individual, it is expected that the results obtained here should be transferable to metapopulations.": This is not obvious, especially given the dependence on update rules which is found throughout the present manuscript. I recommend toning down this sentence and what follows (see the points below).

2b- Line 376: "we have identified amplifiers and suppressors of fixation among the metastar structures presented in [86], see SI Appendix I for more details.": This point is interesting (but see my minor point about SI Appendix I). However, while having mutants that only start in the centre in the model of [86] is something that could be considered experimentally, I do not see how this could happen with spontaneous mutants (e.g. mutants that would appear randomly upon division, as considered in all the rest of the paper). I believe that this is a substantial difference between the result shown in Appendix I and those found in the main text of the manuscript, which assume update rules that constrain mutant appearance.

In fact, it was shown in [86] that when mutants appear upon division, the metastar is an AoS or an SoS depending on migration asymmetry. Given this, I believe that saying that the metastar is an AoF or an SoF can be quite confusing for readers, especially because they might be inclined to assume that mutants appear upon division, as in the rest of the paper. I thus strongly recommend toning down this sentence, and either adding qualifiers such as "the metastar behaves like an AoF or an SoF if mutants are constrained to appear in the centre", or avoiding to state that the metastar can be an AoF or an SoF.

2c- Line 378: "In this way, the results presented in this paper gain empirical relevance and can be tested in experimental settings [82].": The results of [86] are limited to the rare migration regime – as rightly mentioned in Section I of the Supplementary Information, [86] "assumes time scale separation". Unfortunately, in this regime, experiments are slow to the point of not being realistic, at least with current settings. For instance, the experiments of [82] were performed with more frequent migrations. When there is no separation of timescales, the metastar behaves differently. This was recently analysed in Abbara and Bitbol, PNAS Nexus 2(11): pgad392 (2023). I recommend mentioning this for completeness, and toning down the sentence.

2d- The role of deleterious mutations in the evolution of spatially structured population was considered in slightly different models. They impact the exploration of fitness landscape, in particular the crossing of fitness valleys. This was for instance studied recently in the framework of evolutionary graph theory in the recent paper Kuo and Carja, *Genetics* 227(2):iyae055, 2024. I recommend mentioning this for completeness.

Other points:

1- A result often obtained for amplifiers and suppressors of selection is that they do not impact the fixation probability of a neutral mutant. This is related to symmetry properties. It would be nice to comment on why this is not the case for amplifiers or suppressors of fixation. Presumably it is due to the initialization which has a different symmetry.

2- Figure 3 C-D is identical to Figure 2 of Ref. 33 (from the same group), but for another size of graph ($N=8$, instead of 4, 10 and 14). It is fine to show this for comparison with the top panels where the initialization is different, but I recommend mentioning in the figure caption that a similar analysis as Figure 3 C-D was performed in Figure 2 of Ref. 33. This will help the reader.

3- Line 241: "for long-term dB dynamics we expect other (large) regular graphs to have the same steady-state as the complete graph": it would be good if the authors could specify why the condition is that the graph should be regular. Is there a proof of this?

4- Figure 4: Is the distribution of mutant fitnesses assumed to be uniform here (as at line 219)? Eq. 12 is used here, and it requires taking a form for $\rho(f, f')$, unless I missed something. Please specify.

5- Figure 5: "The same fitness discretisation is chosen as in Fig. 3": since Fig. 3 says "see Sec. IIIB", and it is said there that " $f_0 = 0.5; 0.75; 1; 1.25; 1.5; 1.75; 2; 2.25$ and 2.5 ", I would a priori assume that this is what is used here. But I am not sure what this means in this context, since the goal is not to categorize graphs as AoS, SoS, AoF, SoF but rather to calculate the mean steady-state fitness. To me this looks like it would require a distribution of mutant fitnesses, as in Figure 4. I also see that the main text refers to SI Appendix F, which states "The states are labelled with integer values from 0, 1, ..., z. For $1 \leq i \leq z-1$, the fitness of state i is equal to $f_i = f_{min} + i \Delta$ " and moves are between adjacent states. Is this actually what is done in Figure 5? Please clarify these points.

6- Section I and Figure I.1: This exact case, starting from mutants in the centre, was considered and plotted in Figure S3A of the Supplementary Information of [86]. Similarly, Equation I1 is identical to Equation S34 of the Supplementary Information of [86]. Please mention this. Section I may look like a new analysis within the model of [86] to the reader, while it is not the case.

7- I could see no code associated to this manuscript, while simulations were performed for several figures. I strongly recommend publishing the code that allows to reproduce the figures of the manuscript.

Minor points and typos:

1- Paragraph before equation 1: The link between update rule and initialization scheme is nice, and is related to mutants arising upon division. While it is clearly said that "After reproduction, the offspring either resembles its parent with probability $1-\mu$ or is a mutant with probability μ ", it would be helpful to further state that it is this hypothesis that leads to the link between update rule and initialization scheme (if mutants arose e.g. due to stress such as UV radiation, they could a priori appear uniformly). Note that the same point holds for equation 2 in the dB^p case (see below).

2- Paragraph before equation 1: It would be good to remind the reader of the definition of in-degree. Furthermore, for the in-temperature, it would be good to either define the notion or to say that equation 1 a definition of it.

3- Equation 2: it would be good to say that this equation defines the out-temperature T_i^{out} , and to say here too that the link between update rule and initialization is obtained under the assumption that mutants arise at birth.

4- Line 121: "moving to vacant node" -> "moving to a vacant node"

5- Line 152: "Before we proceed, it is important to note that the choice of individual type moving to vacant sites (parent or offspring) only affects the initial mutant placement on the structure. In the absence of mutations, these two choices are equivalent as both the offsprings resemble the parent type.": The placement of these sentences is not clear to me, it would be nice to connect them more explicitly to what comes after or before.

6- Line 224: "This can also be seen by performing the $N \rightarrow \infty$ on the average steady-state": the word "limit" is missing, this should read: "This can also be seen by performing the $N \rightarrow \infty$ limit on the average steady-state".

7- Line 326: "also extends" should read "also extend"

8- Line 330: "it is not only the order of birth and death events, but also the choice of the individual moving to vacant sites significantly impacts the results at the short-term fixation time scale.": This sentence should be amended, e.g. by removing "It

is”.

9- Line 334: “the later” should read “the latter”; “is not good is preventing” should read “is not good at preventing”

10- Supplementary Information: The centred-text format is unusual, and I recommend shifting to justified text. When reading, I had the impression that there were spurious new paragraphs several times because of the centring.

11- Supplementary Information: In Figure B.1, please spell out what the multiple curves with different colours stand for.

(Remarks on code availability)

As mentioned in my review, I could see no code associated to this manuscript, while simulations were performed for several figures. I strongly recommend publishing the code that allows to reproduce the figures of the manuscript.

Reviewer #3

(Remarks to the Author)

First, the authors propose a new justification for the usage of temperature initialization in the dB updating, plus they classify graphs not only to amplifiers and suppressors of selection but also amplifiers and suppressors of fixation, In the amplifier of fixation, a mutant with any fitness has a higher fixation probability than the mutant with the same fitness in the complete graph.

The justification and new classification of amplification are nice and very useful.

These questions are used in section 3, for the short time scales, the results there are mainly results from the literature interpreted in the light of the new questions (AoS/AoF).

Second, the authors examine how different initialization (temperature/uniform) changes the long-term scales, where a small mutation can introduce mutants of different fitness.

In that case, having AoF (which was considered as an amplifier in the literature) might not be beneficial, since the deleterious mutation has a higher chance to fixate as well.

The authors introduce general equations for the density of mutants based on their fitness in their model and then apply them to well-studied graphs and small random graphs.

Overall opinion is as follows:

The questions considered are interesting and the paper presents technically sound analysis and interesting simulation results.

However, there are not enough novel analytical results related to the main questions and different classes of graphs.

In the short-time scale regime, for temperature initialization it presents known results in a new light, but specially for stars there are not novel/new results.

In the long-time scale regime, the paper presents results for complete and regular graphs and stars, which are the most standard

baseline graphs. There are no new graphs where results are presented which significantly outperforms these baseline graphs.

There are a lot of small results, which in some sense dilutes the contributions, and there no standout analytical contribution.

In summary, the analytical results are not novel and exciting for Nature Comm. I would consider this a sound paper relevant for specialized journals (such as Scientific Reports, maybe even for Comm Biology), however, this is below the Nature Comm level.

Comments:

Paragraphs:

In general, I didn't appreciate how the text is broken into paragraphs.

For instance: the paragraph (after line 141) starts talking about the star, then moves to the complete graph, and then continues with the star.

Line 44 should start a new paragraph, otherwise, it hides the definition of SoF and AoF.

Figure 1: the lower part of the graph on 8 vertices is useless in panels A and B.

The main message that the parent or offspring moves can be expressed more straightforwardly without so many vertices.

The first equation (eq4) in section A (star graph) refers to the complete graph. That is confusing and the equation should be in the previous section.

(Also breaking it to paragraphs would help at least a bit)

Figure 3 and section 3. C.

This section is useless. On 8 vertices, there are only 11,117 graphs.

It is better to see how the process behaves on all graphs.

A very well-done figure in this setting is in [1], figure 2.

(The same problem happens in fig5.)

Minor comments:

Eq 3,4 inconsistent notation, ϕ doesn't have the process to which it refers.

Fig2 A, why there is no evaluation (the triangles)?

[1] Tkadlec & al. Population structure determines the tradeoff between fixation probability and fixation time, comm bio.

(Remarks on code availability)

Reviewer #4

(Remarks to the Author)

(Remarks on code availability)

Version 1:

Reviewer comments:

Reviewer #1

(Remarks to the Author)

The authors have adequately addressed all of my comments.

(Remarks on code availability)

Reviewer #2

(Remarks to the Author)

I thank the authors for their comprehensive responses to the reviews, and for their revisions, which substantially improved the clarity of the manuscript. I am fully satisfied with the responses and revisions addressing the questions I asked. In my opinion, this manuscript constitutes an important advance in evolutionary graph theory, and will be of significance to this field. Applications to natural biological situations might be trickier, but this is often the case in this field.

I only have one remaining minor remark, which regards a revised portion of the manuscript.

"By introducing a correlation between initial and final fitness the ruggedness of the landscape can be tuned [61]. Note that the infinite alleles assumption implies that there are no fitness peaks.": I find these two sentences confusing. Indeed, I understand "rugged" precisely as meaning there are several fitness peaks, and would expect "more rugged" to essentially mean that there are more fitness peaks. Checking Ref. [61], I saw that the focus was on correlations there ("Much of the theory of adaptation is concerned with understanding the dynamics on uncorrelated, or "rugged", fitness landscapes."). However, in the literature, "rugged" is often used beyond fully uncorrelated landscapes, e.g. for the Kauffman NK landscapes, and e.g. in "Measuring ruggedness in fitness landscapes", Van Cleve and Weissman, PNAS 112 (24) 7345-7346 (2015). Thus, while I agree that correlations can be tuned and that this is interesting, I recommend removing the mention of "ruggedness" in this context.

(Remarks on code availability)

Reviewer #3

(Remarks to the Author)

The authors in the rebuttal are convincing about the merits of the paper. I recommend acceptance.

I think implications for the evolution on rugged fitness landscapes should be highlighted in the discussion (or introduction).
Figure 1: The changes are good. (Personally I would rotate the left position a little bit, such that the description of the reproduction does not go too much to the right past the midpoint of the arrow.)

Figure 3: Ok, I don't have a problem with the figure by itself, I would only rather see a different one. (Is the star, the main graph present in the plots? I see it as unlikely.) With the change in the abstract, I'm fine with the figure staying.

(Remarks on code availability)

Reviewer #4

(Remarks to the Author)

(Remarks on code availability)

Reviewer #1 (Remarks to the Author):

In the paper "Graph-structured populations elucidate the role of deleterious mutations in long-term evolution" by N.Sharma et al, the authors investigate both short-term and long term evolutionary dynamics on graphs. To this end, they provide a very nice classification of updating and initialization schemes that appear in the Moran process on graphs, and show how different rules lead to very different properties of the system (suppressors/amplifiers of selection, and suppressors/amplifiers of fixation).

As someone working in this area I found this type of classification very useful, and the results fascinating. I have several comments that may help improve clarity of the paper and highlight its impact. These are in no particular order.

Thank you for your positive remarks and the constructive feedback.

-- In the abstract, the authors use the terminology such as death-Birth and Birth-death. I think most people won't know what capitalization stands for. So maybe it is possible to explain it there. And, instead of listing all the results, maybe it is even more important to highlight the important points that are often overlooked (such as that it matters whether birth or death is under selection, and it matters where the mutant is placed).

Agreed, we have now made this implicit terminology more clear.

-- Again, in the abstract, I suggest that more precise language is used: instead of "most random graphs" perhaps "most Erdos-Renyi graphs under 9 nodes".

We agree and we have now changed the sentence.

-- The authors only considered the dB and Bd processes. Can they say something about Db and bD processes? Perhaps there is some symmetry in the patterns.

The work by Kaveh et al. RSOS 2015 and Débarre et al. NatComm 2014 shows that for regular graphs the fixation probabilities under Db updating are equivalent to that under Bd, and dB is equivalent to bD. However, for heterogenous graph this is not true. As an example, the figure 2 of Hadjichrysanthou et al. Dynamics Games and Evolution 2011 shows that fixation probability for the star graph under Db is not equivalent to the one under Bd updating. This indicates that there is unfortunately not the kind of symmetry one would hope for.

-- On page 4, the derivations correspond to directed graphs, but since the paper only deals with undirected graphs, this can be significantly shortened (to reduce confusion).

We agree. However, the derivation does not shorten significantly when focusing solely on the undirected graphs. To avoid confusion, we have removed comments on the general case of weighted graphs to put more emphasis on the undirected graphs.

-- Type: l.105: rules rules (repetition)

Thanks.

-- The authors only consider the regime of sequential fixation. Would it be interesting to include somewhat higher mutation rates and see if the results change under stochastic tunneling?

It is definitely interesting to look into higher mutation rates and the process of valley crossing, and we are currently pursuing this direction. However, we think that this is a topic for a separate manuscript. We have now mentioned related work by Komarova et al. JRSI 2014 and Kuo & Carja Genetics 2024 in the Discussion proposing higher mutation supply as an interesting future direction.

-- Fig.2 (and other similar graphics): the graphics showing star graphs with the central node and one of the leaf nodes circled are confusing. As far as I understand, the authors are not comparing the probability of fixation of mutants that are put in the center vs the periphery. Instead, they are comparing the initial placement schemes denoted by "o" and "p", which result in a preferential placement in the center or in the periphery. Maybe this could be explained in the figure caption.

We agree, it can be a potential source of confusion. To clarify this, in the caption we have mentioned that for large population sizes, comparing the update rules "o" and "p" corresponds to the initial mutant placement on the center and leaf for the star graph.

--In the Discussion section, some experiments are cited, for which these theories could be relevant. It would be great to give some more details of what biological systems are those?

We have added a sentence briefly describing the experiment of Ref.[8], and added another recent experimental reference demonstrating decreased selection and genetic drift in a structured environment.

Reviewer #2 (Remarks to the Author):

This manuscript presents a study of spatially structured populations on graphs, in the framework of evolutionary graph theory. It sheds light on new possible impacts of graph structures under a new update rule. In particular, this update rule gives rise to suppressors of fixation (SoF), which decrease the fixation probability of all mutants compared to a well-mixed population. Interestingly, populations on these graphs can attain higher fitness in the long-term.

This manuscript is extremely complete, as it considers various aspects of a new update rule, including its relationship to the mutant initialization, its impact on fixation probability in simple graphs, but also in random graphs, and a detailed analysis of the impact on long-term fitness. I particularly appreciated the discussion of the relationship between update rules and initialization schemes under the assumption that mutations occur at division.

It is also scientifically sound and very well-presented, and overall clear and easy to read. In my opinion, it constitutes an important advance in evolutionary graph theory.

However, I found some issues regarding the link to natural biological situations, and to other models and to experiments. They are spelled out in my major points below.

Other important points regard some clarifications and links to published works. They are detailed in my other points below. Finally, there are a few minor points and typos that I recommend correcting.

Thank you for your constructive feedback and kind words.

Major points:

1- While this manuscript presents a theoretical study, connecting with natural biological situations is important. At the moment, there are some points that are unclear or potentially misleading about this. I recommend clarifying them. In particular:

1a- Line 100: "lizards and snakes": it is not clear at all that the model presented here would be valid for them. First, these organisms are diploids that reproduce sexually, which is not modelled here. Second, they move actively in continuous space, meaning that a static graph may not be an adequate representation of spatial structure in their case. It would be good to either clarify and mitigate this, or to remove this example.

We fully agree, and have removed the lizard-snake example.

1b- Line 105: "microbial and somatic cell populations, where parent and offspring cannot necessarily be distinguished (...) there is no reason why the daughter
111 cell identical to the mother cell cannot move instead": These are very good points. But don't they argue for studying update rules where either the parent or the offspring may move, e.g. with 50% probability, rather than analysing the two extreme cases where only one of them moves? It would be good to discuss this.

Thank you for the great suggestion! In Appendix B, we have added a new analysis on what we call the ω dB process. In a given update step, with probability ω , the offspring moves and otherwise, the parent moves. We find that in the limit of an infinitely large star graph, for any $\omega < 1$, the probability to fix deleterious mutants is non-zero.

Moreover, when subjected to long-term evolution, the steady-state fitness statistics of a very large star graph for any non-zero ω values converge to the statistics for the $\omega = 0$ case (only parent moving).

This shows that a very small proportion of parent moving is sufficient to significantly affect the long-term evolution.

We now mention these results in the main text well.

1c- Line 219: "we assume that the mutant's fitness is sampled from a uniform distribution": It would be valuable to say a bit more about the realism of this assumption.

We have considerably expanded the motivation of the model in the introductory part of Section IV. We now explain that the choice of a continuous random mutant fitness corresponds to an infinite alleles setting. For the House-of-Cards model, where the mutant fitness is independent of the parental fitness, we have added an explicit expression for the stationary fitness distribution [Eq.(14)]. This should make it clear that the specific choice of a uniform mutant fitness distribution in the subsequent sections was made for convenience only and does not affect the results in a significant way. Note also that the results obtained for Fig. 5 considered nearest neighbour hopping in the (discretized) fitness space. The results for the star graph are qualitatively the same as for uniform mutant fitness distribution.

2- Similarly, the link with other models and with experiments can be improved, in particular:

2a- Line 374: "While our work focuses on the setting of one-node-one-individual, it is expected that the results obtained here should be transferable to metapopulations.": This is not obvious, especially given the dependence on update rules which is found throughout the present manuscript. I recommend toning down this sentence and what follows (see the points below).

True. While our work focuses on the setting of one-node-one-individual, extending to network structured metapopulations is an important future direction. For some update rules, it is known that results carry over to this case, but this is a largely unexplored area (Yagoobi et al., 2023).

We have now modified the text accordingly.

2b- Line 376: "we have identified amplifiers and suppressors of fixation among the metastar structures presented in [86], see SI Appendix I for more details.": This point is interesting (but see my minor point about SI Appendix I). However, while having mutants that only start in the centre in the model of [86] is something that could be considered experimentally, I do not see how this could happen with spontaneous mutants (e.g. mutants that would appear randomly upon division, as considered in all the rest of the paper). I believe that this is a substantial difference between the result shown in Appendix I and those found in the main text of the manuscript, which assume update rules that constrain mutant appearance.

In fact, it was shown in [86] that when mutants appear upon division, the metastar is an AoS or an SoS depending on migration asymmetry. Given this, I believe that saying that the metastar is an AoF or an SoF can be quite confusing for readers, especially because they might be inclined to assume that mutants appear upon division, as in the rest of the paper.

I thus strongly recommend toning down this sentence, and either adding qualifiers such as "the metastar behaves like an AoF or an SoF if mutants are constrained to appear in the centre", or avoiding to state that the metastar can be an AoF or an SoF.

Agreed. We have now modified the above mentioned sentence to, "...provided the central deme is initialized with mutants, the metastar structure presented in [72] behaves like an amplifier of fixation or a suppressor of fixation depending on the migration asymmetry between demes..."

2c- Line 378: "In this way, the results presented in this paper gain empirical relevance and can be tested in experimental settings [82].": The results of [86] are limited to the rare migration regime – as rightly mentioned in Section I of the Supplementary Information, [86] "assumes time scale separation". Unfortunately, in this regime, experiments are slow to the point of not being realistic, at least with current settings. For instance, the experiments of [82] were performed with more frequent migrations. When there is no separation of timescales, the metastar behaves differently. This was recently analysed in Abbara and Bitbol, PNAS Nexus 2(11): pgad392 (2023). I recommend mentioning this for completeness, and toning down the sentence.

Agreed. Thank you for the suggestion, we agree and have modified the text accordingly. "In this way, the results presented in this paper can gain empirical relevance [Abbara et al. 2024]. There are experimental challenges associated with the separation of the time-scales of mutation and fixation as assumed here, and the metastar is predicted to behave differently when this assumption is relaxed [Abbara and Bitbol, 2023]. Nonetheless, our work indicates that examining

the role of the deleterious mutant regime for structured metapopulations is an important future direction.”

2d- The role of deleterious mutations in the evolution of spatially structured population was considered in slightly different models. They impact the exploration of fitness landscape, in particular the crossing of fitness valleys. This was for instance studied recently in the framework of evolutionary graph theory in the recent paper Kuo and Carja, *Genetics* 227(2):iyae055, 2024. I recommend mentioning this for completeness.

Thank you for the recommendation and pointing us to the reference. We have now covered this in the discussion.

Other points:

1- A result often obtained for amplifiers and suppressors of selection is that they do not impact the fixation probability of a neutral mutant. This is related to symmetry properties. It would be nice to comment on why this is not the case for amplifiers or suppressors of fixation. Presumably it is due to the initialization which has a different symmetry.

Even under neutrality, for an arbitrary graph, a mutant can have different fixation probability depending on the node it appears at. The sum of those probabilities is equal to 1 as one of the individuals will eventually fix. This way, the uniform initialised fixation probability on any graph is $1/N$ after averaging over all nodes.

When it comes to temperature initialization, the averaging of individual node fixation probabilities is not equal to $1/N$. As a result, the fixation probability is not equal to $1/N$ for heterogenous graphs. We have now cited Allen et al. *PlosCB* 2015 in the section III B emphasizing these points.

2- Figure 3 C-D is identical to Figure 2 of Ref. 33 (from the same group), but for another size of graph ($N=8$, instead of 4, 10 and 14). It is fine to show this for comparison with the top panels where the initialization is different, but I recommend mentioning in the figure caption that a similar analysis as Figure 3 C-D was performed in Figure 2 of Ref. 33. This will help the reader.

Thank you, we covered this in the caption.

3- Line 241: “for long-term dB dynamics we expect other (large) regular graphs to have the same steady-state as the complete graph”: it would be good if the authors could specify why the condition is that the graph should be regular. Is there a proof of this?

We do not have a formal proof. It is more of a hypothesis based on the results we have for long-term fitness statistics for the cycle and lattice graph. It could be a non-regular graph, but we are not sure.

We have now mentioned in the main text more prominently that this is only a hypothesis.

4- Figure 4: Is the distribution of mutant fitnesses is assumed to be uniform here (as at line 219)? Eq. 12 is used here, and it requires taking a form for $\rho(f,f')$, unless I missed something. Please specify.

That's right. The mutant fitness distribution is uniform here, we have now clarified this.

5- Figure 5: "The same fitness discretisation is chosen as in Fig. 3": since Fig. 3 says "see Sec. IIIB", and it is said there that " $f_0 = 0.5; 0.75; 1; 1.25; 1.5; 1.75; 2; 2.25$ and 2.5 ", I would a priori assume that this is what is used here. But I am not sure what this means in this context, since the goal is not to categorize graphs as AoS, SoS, AoF, SoF but rather to calculate the mean steady-state fitness. To me this looks like it would require a distribution of mutant fitnesses, as in Figure 4. I also see that the main text refers to SI Appendix F, which states "The states are labelled with integer values from $0, 1, \dots, z$. For $1 \leq i \leq z-1$, the fitness of state i is equal to $f_i = f_{\min} + i \Delta$ " and moves are between adjacent states. Is this actually what is done in Figure 5? Please clarify these points.

Yes, thank you pointing that out. We have considered nearest neighbour mutant jumps in Fig 5. Additionally, this is an example of using a mutant fitness distribution other than the uniform distribution. This figure suggests that the steady-state results for the star graph are qualitatively robust to different choices of mutational fitness distribution. The same is expected for an arbitrary graph.

6- Section I and Figure I.1: This exact case, starting from mutants in the centre, was considered and plotted in Figure S3A of the Supplementary Information of [86]. Similarly, Equation I1 is identical to Equation S34 of the Supplementary Information of [86]. Please mention this. Section I may look like a new analysis within the model of [86] to the reader, while it is not the case.

We agree and have now cited [Marrec et al 2021] at multiple places to indicate that the results are taken from Marrec et al. 2021 and that they are not ours.

7- I could see no code associated to this manuscript, while simulations were performed for several figures. I strongly recommend publishing the code that allows to reproduce the figures of the manuscript.

We apologize for this shortcoming. We have now made our codes available.

Minor points and typos:

1- Paragraph before equation 1: The link between update rule and initialization scheme is nice, and is related to mutants arising upon division. While it is clearly said that "After reproduction, the offspring either resembles its parent with probability $1-\mu$ or is a mutant with probability μ ", it would be helpful to further state that it is this hypothesis that leads to the link between update rule and initialization scheme (if mutants arose e.g. due to stress such as UV radiation, they could a priori appear uniformly). Note that the same point holds for equation 2 in the dB^p case (see below).

Thank you for pointing this out. We have mentioned it in the first paragraph of section II that we work with mutations coupled to birth.

We also cover this in the beginning of the discussion.

2- Paragraph before equation 1: It would be good to remind the reader of the definition of in-

degree. Furthermore, for the in-temperature, it would be good to either define the notion or to say that equation 1 a definition of it.

Thank you.

Since we work with undirected graphs, in-degree and out-degree is the same. We have changed in/out-degree to degree.

3- Equation 2: it would be good to say that this equation defines the out-temperature T_i^{out} , and to say here too that the link between update rule and initialization is obtained under the assumption that mutants arise at birth.

We have now made this clear in the beginning of this section.

4- Line 121: "moving to vacant node" -> "moving to a vacant node"

Done. Thank you.

5- Line 152: "Before we proceed, it is important to note that the choice of individual type moving to vacant sites (parent or offspring) only affects the initial mutant placement on the structure. In the absence of mutations, these two choices are equivalent as both the offsprings resemble the parent type.": The placement of these sentences is not clear to me, it would be nice to connect them more explicitly to what comes after or before.

Thank you for pointing this out. We have now modified the text.

6- Line 224: "This can also be seen by performing the $N \rightarrow \infty$ on the average steady-state": the word "limit" is missing, this should read: "This can also be seen by performing the $N \rightarrow \infty$ limit on the average steady-state".

Done.

7- Line 326: "also extends" should read "also extend"

Done.

8- Line 330: "it is not only the order of birth and death events, but also the choice of the individual moving to vacant sites significantly impacts the results at the short-term fixation time scale.": This sentence should be amended, e.g. by removing "It is".

Done.

9- Line 334: "the later" should read "the latter"; "is not good is preventing" should read "is not good at preventing"

Done.

10- Supplementary Information: The centred-text format is unusual, and I recommend shifting to justified text. When reading, I had the impression that there were spurious new paragraphs several times because of the centring.

Done.

11- Supplementary Information: In Figure B.1, please spell out what the multiple curves with different colours stand for.

Done.

Reviewer #2 (Remarks on code availability):

As mentioned in my review, I could see no code associated to this manuscript, while simulations were performed for several figures. I strongly recommend publishing the code that allows to reproduce the figures of the manuscript.

We apologize, we fully agree that the code must be shared with the publication.

Reviewer #3 (Remarks to the Author):

First, the authors propose a new justification for the usage of temperature initialization in the dB updating, plus they classify graphs not only to amplifiers and suppressors of selection but also amplifiers and suppressors of fixation. In the amplifier of fixation, a mutant with any fitness has a higher fixation probability than the mutant with the same fitness in the complete graph. The justification and new classification of amplification are nice and very useful. These questions are used in section 3, for the short time scales, the results there are mainly results from the literature interpreted in the light of the new questions (AoS/AoF).

Thank you for these positive remarks.

Second, the authors examine how different initialization (temperature/uniform) changes the long-term scales, where a small mutation can introduce mutants of different fitness. In that case, having AoF (which was considered as an amplifier in the literature) might not be beneficial, since the deleterious mutation has a higher chance to fixate as well. The authors introduce general equations for the density of mutants based on their fitness in their model and then apply them to well-studied graphs and small random graphs.

Overall opinion is as follows:

The questions considered are interesting and the paper presents technically sound analysis and interesting simulation results.

However, there are not enough novel analytical results related to the main questions and different classes of graphs.

In the short-time scale regime, for temperature initialization it presents known results in a new light, but specially for stars there are not novel/new results.

We understand that for the short-time scale part of the manuscript, some of the results were known previously. Accordingly, we had cited the respective references in the main text. With that said, the following results considerably extend the scope of standard evolutionary graph theory:

1. **Relation between individual-type moving and mutant initialization:** With the choice of individual moving, parent or offspring, uniform and temperature mutant initializations come out naturally from the update rule. Moreover, the parent moving dB rule allows to justify temperature initialized dB updating which earlier in the literature was suggested to not exist [Tkadlec, Pavlogiannis et al. PlosCB 2020, McAvoy et al. PlosCB 2021].
2. **Amplifier of fixation:** By allowing parent individuals to move, we have found a new category of graphs for dB updating, namely amplifiers of fixation-- structures with a higher fixation probability for mutants regardless of their fitness values compared to well-mixed population. The star graph has a non-zero fixation probability for deleterious mutations even in the limit of infinite population size. This contradicts the established view that the probability to fix deleterious mutants always goes to zero for large population sizes.

In the long-time scale regime, the paper presents results for complete and regular graphs and stars, which are the most standard baseline graphs. There are no new graphs where results are presented which significantly outperforms these baseline graphs.

There are a lot of small results, which in some sense dilutes the contributions, and there no standout analytical contribution.

In summary, the analytical results are not novel and exciting for Nature Comm. I would consider this a sound paper relevant for specialized journals (such as Scientific Reports, maybe even for Comm Biology), however, this is below the Nature Comm level.

Thank you for your feedback.

One of the questions we address in this manuscript is, "Are fixation probabilities for beneficial mutations sufficient to explain long-term evolution?" The short answer is no. Below, we outline the contributions our work makes to evolutionary graph theory and evolutionary microbial biology.

1. **Expanding the focus beyond beneficial mutations:** Most studies in evolutionary graph theory focus on amplifiers of selection, placing emphasis on the beneficial mutation regime. Our work shows that the deleterious mutation regime can be just as important as the beneficial one and, therefore, warrants equal consideration. While it is generally assumed that structures that effectively fix beneficial mutants will achieve higher fitness in the long-term evolution, we demonstrate that this view is incomplete. Specifically, suppressors of fixation—structures that are less effective at fixing beneficial mutations but better at removing deleterious ones—can achieve higher fitness than not only well-mixed populations but also amplifiers of selection. Conversely, amplifiers of fixation—structures better at fixing beneficial mutations but less effective at eliminating deleterious ones—can have lower fitness than both well-mixed populations and suppressors of selection. These findings emphasize the need to account for the deleterious mutation regime in structured populations.
2. **Implications for evolution on rugged fitness landscapes:** Beyond evolutionary graph theory, adaptation on fitness landscapes is a central topic in theoretical population genetics. Although fixing deleterious mutations can aid in navigating rugged fitness landscapes, well-mixed populations have limited ability to do so especially when the population size is large. The amplifiers of fixation identified in our study are more likely than well-mixed populations to cross fitness valleys, and may thus enable better

navigation of rugged landscapes (work in preparation). Evolutionary graph theory provides a platform to design structures with tailored fixation probabilities for both beneficial and deleterious mutations, offering a means to control and navigate evolutionary processes. And to achieve that, it is crucial to also focus on the deleterious mutant regime.

Comments:

Paragraphs:

In general, I didn't appreciate how the text is broken into paragraphs.

For instance: the paragraph (after line 141) starts talking about the star, then moves to the complete graph, and then continues with the star.

Line 44 should start a new paragraph, otherwise, it hides the definition of SoF and AoF.

Thank you for the suggestion, we have now rearranged the text.

Figure 1: the lower part of the graph on 8 vertices is useless in panels A and B.

The main message that the parent or offspring moves can be expressed more straightforwardly without so many vertices.

Thank you. We agree that the point can be made with smaller graphs, but we also understand that some readers are not used to the associated higher level of abstraction. Moreover, we need the lower part of the figures to emphasize that we start with a monomorphic population and illustrate the choice of update affecting the mutant initialization. Following the reviewer's suggestion, we have reduced the size of the graphs.

The first equation (eq4) in section A (star graph) refers to the complete graph. That is confusing and the equation should be in the previous section.

(Also breaking it to paragraphs would help at least a bit)

Thank you. We have rearranged the text now.

Figure 3 and section 3. C.

This section is useless. On 8 vertices, there are only 11,117 graphs.

It is better to see how the process behaves on all graphs.

A very well-done figure in this setting is in [1], figure 2.

(The same problem happens in fig5.)

Thank you. Here, we politely disagree. In sec 3 C, we compare our results with the previous work by Hindersin and Traulsen Plos CB 2015. We want to make the point that it is easy to construct an amplifier/suppressor of fixation without going into the question of "how many of all graphs are amplifiers/suppressors of fixation". A detailed study as you suggest (similar to Möller et al. 2020) is certainly an important future direction.

However, in Fig. 2 of the reference suggested by the reviewer, the plots are made assuming one mutant fitness value only ($f' > 1$). The question we are interested in requires comparing fixation probability profiles, rather than the fixation probability for a given fitness value. Then, one could show the fraction of amplifiers of fixation, suppressors of fixation, etc. in a pie chart for a fixed population size. However, from discussions with colleagues in the field we understand that the message that it is easy to construct networks of certain categories is important.

Minor comments:

Eq 3,4 inconsistent notation, ϕ doesn't have the process to which it refers.

We have now made changes to Eq. 4.

For Eq. 3 we think it is good to not have dB or Bd on the notation.

Fig2 A, why there is no evaluation (the triangles)?

Fig 2 A is a known result, so we did not do simulations to confirm the analytics.

We have now clarified this in the caption.

[1] Tkadlec & al. Population structure determines the tradeoff between fixation probability and fixation time, comm bio.

Reviewer #4 (Remarks to the Author):

Thank you for your constructive feedback and kind words.

Reviewer #1

The authors have adequately addressed all of my comments.

We thank you for your valuable suggestions and feedback.

Reviewer #2

I thank the authors for their comprehensive responses to the reviews, and for their revisions, which substantially improved the clarity of the manuscript. I am fully satisfied with the responses and revisions addressing the questions I asked. In my opinion, this manuscript constitutes an important advance in evolutionary graph theory, and will be of significance to this field. Applications to natural biological situations might be trickier, but this is often the case in this field.

Thank you for the kind words.

I only have one remaining minor remark, which regards a revised portion of the manuscript. "By introducing a correlation between initial and final fitness the ruggedness of the landscape can be tuned [61]. Note that the infinite alleles assumption implies that there are no fitness peaks.": I find these two sentences confusing. Indeed, I understand "rugged" precisely as meaning there are several fitness peaks, and would expect "more rugged" to essentially mean that there are more fitness peaks. Checking Ref. [61], I saw that the focus was on correlations there ("Much of the theory of adaptation is concerned with understanding the dynamics on uncorrelated, or "rugged", fitness landscapes."). However, in the literature, "rugged" is often used beyond fully uncorrelated landscapes, e.g. for the Kauffman NK landscapes, and e.g. in "Measuring ruggedness in fitness landscapes", Van Cleve and Weissman, PNAS 112 (24) 7345-7346 (2015). Thus, while I agree that correlations can be tuned and that this is interesting, I recommend removing the mention of "ruggedness" in this context.

Thank you, we agree.

We have modified the respective text, and removed the entire sentence that contained the word "ruggedness".

Reviewer #3:

The authors in the rebuttal are convincing about the merits of the paper. I recommend acceptance.

Thank you for the feedback.

I think implications for the evolution on rugged fitness landscapes should be highlighted in the discussion (or introduction).

As per reviewer 2 suggestion as well, in discussion (line 377-378) we have stated the connection of our work to valley crossing problems on rugged fitness landscapes.

Figure 1: The changes are good. (Personally I would rotate the left position a little bit, such that

the description of the reproduction does not go too much to the right past the midpoint of the arrow.)

Thank you for the suggestion, we have modified the figure accordingly.

Figure 3: Ok, I don't have a problem with the figure by itself, I would only rather see a different one.(Is the star, the main graph present in the plots? I see it as unlikely.)With the change in the abstract, I'm fine with the figure staying.

Thank you. Regarding the question, we also see it is unlikely that a star would appear in the plot. However, this is not particularly relevant, as the main purpose of the plot is to convey the likelihood of a randomly generated graph to belong to a specific graph category.

Reviewer #4:

Thank you for the suggestions and constructive feedback.